# Inflammation as a Prognostic Marker in Cardiovascular Kidney Metabolic Syndrome: A Systematic Review

**DOI:** 10.3390/ijms27010134

**Published:** 2025-12-22

**Authors:** Sihle E. Mabhida, Haskly Mokoena, Mamakase G. Sello, Cindy George, Musawenkosi Ndlovu, Thabsile Mabi, Sisa Martins, Innocent S. Ndlovu, Onyemaechi Azu, André P. Kengne, Zandile J. Mchiza

**Affiliations:** 1Non-Communicable Diseases Research Unit, South African Medical Research Council, Tygerberg, Cape Town 7505, South Africa; cindy.george@mrc.ac.za (C.G.); musawenkosi.ndlovu@mrc.ac.za (M.N.); thabsile.mabi@mrc.ac.za (T.M.); sisa.martins@mrc.ac.za (S.M.); innocent.ndlovu@mrc.ac.za (I.S.N.); andre.kengne@mrc.ac.za (A.P.K.); zandile.mchiza@mrc.ac.za (Z.J.M.); 2Department of Physiology and Environmental Health, University of Limpopo, Sovenga, Polokwane 0727, South Africa; haskly.mokoena@ul.ac.za; 3Department of Medical Biosciences, University of the Western Cape, Bellville, Cape Town 7535, South Africa; desmasello@gmail.com (M.G.S.); oazu@uwc.ac.za (O.A.); 4Department of Nutrition, Faculty of Health Sciences, National University of Lesotho, Maseru 100, Lesotho; 5Department of Medicine, University of Cape Town, Cape Town 7700, South Africa; 6African Population and Health Research Center, Nairobi P.O. Box 10787-00100, Kenya

**Keywords:** CKMS, inflammation, prognosis, biomarkers, mortality, systemic inflammation

## Abstract

Cardiovascular–kidney–metabolic syndrome (CKMS) represents the intricate interconnection of cardiovascular, kidney, and metabolic disorders, with systemic inflammation now recognized as a key driver of both pathogenesis and prognosis. This systematic review aimed to synthesize current evidence on the prognostic value of inflammatory biomarkers in individuals with CKMS. A systematic search of PubMed, Embase, CINAHL, Web of Science, and Scopus were conducted to identify studies published between 1 January 2024 and 30 June 2025, following the recognition of CKMS as a distinct syndrome in December 2023. Eligible studies included adults (aged ≥ 18 years) with CKMS, that assesses one or more inflammatory markers and reported prognostic outcomes such as mortality or disease progression. Data extracted included study characteristics, biomarker types, outcome measures, and key findings. In addition to longitudinal cohorts, we included a small number of cross-sectional studies and treated them as association (non-prognostic) evidence analyzed in a separate stream from prognostic cohorts. Risk of bias was evaluated using the Quality in Prognostic Studies (QUIPS) tool. Due to considerable variability in prognostic outcomes, follow-up durations, and inflammatory indices, a meta-analysis was not feasible. Instead, a narrative synthesis was undertaken to summarize the evidence, identify consistent associations, and emphasize the need for standardized approaches and biomarker validation in future CKMS research. Analysis was conducted in line with the SWiM guidelines. Thirteen studies (*n* = 13) comprising 282,016 participants (100,590 males; 97,295 females) were included from 1404 initial records. Five of the studies were cross-sectional, providing information on associations rather than prognostic outcomes. Most were large-scale cohort studies conducted in the USA and China. Frequently assessed biomarkers included systemic inflammatory response index (SIRI), systemic immune-inflammation index (SII), high-sensitivity C-reactive protein to high-density lipoprotein cholesterol ratio (hs-CRP/HDL-C), dietary inflammatory index (DII), and triglyceride–glucose (TyG) index. Elevated levels of these biomarkers were consistently associated with higher risk of all-cause and cardiovascular mortality, CKMS progression, and adverse metabolic outcomes. This review highlights systemic inflammation as a critical and associated marker of CKMS prognosis. Inflammatory biomarkers may assist in hypothesis generation, but clinical utility remains to be established pending standardized adjustment and external validation. Because CKMS has only recently been operationalized, we limited inclusion to studies published from 1 January 2024 onward, enhancing definitional comparability but narrowing the evidence base and potentially emphasizing early-adopter regions (predominantly the U.S. and China). Accordingly, these findings should be interpreted as early signals that require replication in diverse settings and confirmation through longitudinal and interventional studies to inform integrative CKMS management strategies. Across observational studies, the certainty of evidence is low to moderate due to indirectness and imprecision; findings should be treated as associational signals pending external validation.

## 1. Introduction

Cardiovascular–kidney–metabolic syndrome (CKMS), formally recognized in December 2023, is a complex condition that links cardiovascular disease, chronic kidney disease, and metabolic disorders such as diabetes and obesity. Affecting millions globally, it is increasingly recognized as a major contributor of premature mortality [1]. In the current era of non-communicable diseases (NCDs), deaths rarely result from a single condition. Instead, most arise from interconnected, multifactorial syndromes such as CKMS [1,2,3]. CKMS arises from the interplay of biological, lifestyle, and environmental factors, with chronic low-grade inflammation serving as a central mechanism that unifies metabolic dysfunction, cardiovascular disease, and renal impairment [3,4]. This integrative perspective reflects the evolving understanding that these diseases rarely occur in isolation but rather co-exist and synergistically exacerbate each other, significantly amplifying both morbidity and mortality [4]. Emerging evidence highlights the critical role of inflammation in the pathophysiology of CKMS [5,6].

Classical inflammatory biomarkers such as C-reactive protein (CRP), tumor necrosis factor-alpha (TNF-α), interleukin-6 (IL-6), and interleukin-1 beta (IL-1β) are well-established indicators of systemic inflammation and are commonly used to assess disease presence and severity in metabolic and cardiorenal disorders [5,7,8] recent years, novel inflammatory indices have gained attention for their robust prognostic and predictive value. These include the systemic inflammatory response index (SIRI), systemic immune-inflammation index (SII), and triglyceride–glucose (TyG)-related indices, which provide more nuanced insights into inflammatory status and disease outcomes [7,9,10,11]. Notably, SIRI has been shown to outperform SII in predicting all-cause and cardiovascular mortality, particularly in obese populations. Elevated SIRI levels have also been associated with an increased risk of mortality in patients with CKD, particularly during the early stages of disease progression [7,12,13]. Additionally, the dietary inflammatory index (DII) has emerged as a useful tool for assessing the inflammatory potential of an individual’s diet. Studies have demonstrated that higher DII scores are significantly associated with increased risk of the metabolic syndrome, systemic inflammation, and renal insufficiency, highlighting the role of diet as a modifiable contributor to CKMS [14]. This growing body of evidence underscores the potential for inflammatory biomarkers and composite indices to serve as powerful tools for early risk stratification, personalized treatment, and disease monitoring in individuals with CKMS.

Despite the well-established role of inflammation in NCDs [7,9], there is a lack of consolidated evidence examining its contribution across the full spectrum of CKMS. Existing studies are fragmented across disciplines, often focusing on isolated components such as cardiovascular or CKD-related outcomes [7,9,12], without considering their interconnected nature. Furthermore, although novel inflammatory indices and dietary inflammation scores show promise for risk prediction and clinical decision-making, their relevance to CKMS remains underexplored in a comprehensive, systematic manner. This systematic review aims to bridge this gap by synthesizing current human studies to elucidate the role and prognostic significance of inflammatory markers in the development and progression of CKMS.

## 2. Methods

### 2.1. Study Design

This systematic review was conducted and reported in accordance with the Preferred Reporting Items for systematic reviews and meta-analysis (PRISMA) 2020 and the guide to systematic review and meta-analysis of prognostic factor studies [15]. The systematic review was further structured following PICOTS (Population, Index prognostic factor, Comparator prognostic factor, Outcomes, Timing and Setting) in accordance with the CHARMS checklist:

**Population:** Adults (≥18 years) diagnosed with CKMS or its clinical components

**Index prognostic factor:** Inflammatory indicators or indices (e.g., SIRI, SII, hs-CRP/HDL-C ratio, DII, TyG indices).

**Comparator prognostic factor:** Individuals with lower levels or normal levels of inflammatory markers.

**Outcome:** Prognostic endpoints including all-cause mortality, cardiovascular mortality, CKMS progression, or composite adverse clinical outcomes.

**Timing:** No restriction of time

**Setting:** The study includes patients with CKM (different stages).

This review aimed to comprehensively evaluate the role and prognostic value of inflammatory indices, including SIRI, SII, hs-CRP/HDL-C ratio, DII, and TyG indices, in the context of CKMS. This review was retrospectively registered in the International Prospective Register of Systematic Reviews (PROSPERO, registration no: CRD420251131929) on 22 August 2025. The public record URL is provided in Appendix A

### 2.2. Information Sources and Electronic Database Search Strategy

Key electronic databases of clinical and public health research, including PubMed, Embase, CINAHL, Web of Science, and Scopus, were searched to identify studies examining the association between inflammatory indices and CKMS. Two independent assessors (S.E.M, and M.G.S), with guidance from a librarian, performed the electronic database search using key terms and Medical Subject Headings (MeSH), and applied filters to identify published studies. The review limited its search to studies published between 1 January 2024 and the final search was run on 30 June 2025 (2024–2025) for all databases. (Table 1), following the formal recognition of CKMS by the American Heart Association (AHA) in December 2023 [1]. Pre-2024 studies typically operationalized overlapping, but non-identical, constructs (e.g., “cardio-renal-metabolic” clusters, MetS + CKD) with heterogeneous inclusion criteria and outcomes. Mixing these with post-2023 CKMS definitions risks conceptual misclassification and threatens internal validity of any synthesis (e.g., incomparable staging, outcome baselines, and codes). Restricting to the post-2023 era therefore improves construct comparability across studies, enables consistent staging-based subgrouping, and reduces bias from definitional drift, while we transparently quantify the trade-off in the Limitations Section. To maximize the retrieval of relevant studies, key terms and MeSH, along with their synonyms, developed in alignment with PICOTS components, were search in electronic databases using Boolean operators as follows:

**Inflammatory Biomarkers**: “C-reactive protein” OR “CRP” OR “Systemic Immune-Inflammation Index” OR “SII” OR “Systemic Inflammation Response Index” OR “SIRI” OR “Interleukin” OR “TNF-alpha” OR “Dietary Inflammatory Index” OR “DII” OR “Triglyceride-glucose index” OR “TyG index” OR “inflammatory markers”

**Disease Components:** “cardiovascular disease” OR “chronic kidney disease” OR “CKD” OR “hypertension” OR “type 2 diabetes” OR “insulin resistance” OR “obesity” OR “metabolic syndrome” OR “cardiovascular kidney metabolic syndrome” OR “CKMS”

**Prognostic Indicators:** “mortality” OR “disease progression” OR “outcomes” OR “risk prediction” OR “all-cause mortality” OR “cardiovascular mortality” OR “renal outcome” OR “survival analysis” OR “prognosis”

**Study Design Filters:** “cohort” OR “longitudinal” OR “prospective” OR “retrospective” OR “cross-sectional”

The full search strategy is provided in Appendix A. Reference lists of the included studies and relevant reviews were also hand-searched to identify additional eligible studies. All retrieved articles deemed eligible were imported into EndNote for de-duplication. Titles and abstracts were independently screened by two reviewers (S.E.M, and M.G.S), followed by the full-text assessment of all prospective articles. Disagreements were resolved through discussion or adjudication by a third reviewer (HM).

**Table 1 ijms-27-00134-t001:** Inclusion and Exclusion.

Inclusion	Exclusion
Post-2024 CKMS Studies	Pre-2024 Studies (MetS, CKD + DM, cardio-renal-metabolic disorders)
Human adults (≥18 years) with CKMS	Pediatric populations (<18 years); animal or pre-clinical studies
Inflammation-related markers/indices (e.g., SIRI, SII, hs-CRP/HDL-C, DII, TyG indices) measured at baseline	Studies not reporting inflammatory markers/indices
Prognostic endpoints: all-cause or cardiovascular/renal mortality, CKMS progression, or composite adverse outcomes	Non-prognostic outcomes only (e.g., prevalence without outcomes)
Original, peer-reviewed research	Reviews, editorials, commentaries; protocols/abstracts without full data
Observational designs: prospective or retrospective cohorts; cross-sectional (hypothesis-generating only)	Randomized controlled trials without prognostic analyses; case reports/series without comparative data
No language restrictions; titles/abstracts screened via automated translation; full texts translated where feasible	Full-text translation to English not feasible or unreliable

### 2.3. Eligibility Criteria

This review included original, peer-reviewed studies involving human participants with CKMS aged ≥ 18 years, published post 2024 (Table 1). There was no language restrictions applied unless full-text translation into English was not feasible. Titles and abstracts in non-English languages were screened using automated translation tools (Google Translate; Google LLC, Mountain View, CA, USA), and potentially eligible full-text articles were translated using institutional or publicly available resources. We included observational designs, prospective/retrospective cohorts (prognostic), and cross-sectional studies (non-prognostic). Although prognosis is best assessed with longitudinal data, we included a small number of cross-sectional studies and interpreted them separately as hypothesis-generating only, not as evidence of prognosis. This preserves methodological rigor by distinguishing simple associations from temporal relationships. Studies were excluded if they were pre-clinical, non-original (reviews, editorials), focused on pediatric populations, or did not investigate associations between inflammatory indices and CKMS outcomes. We restricted eligibility to studies in human adults and excluded reviews and other non-original articles (Table 1). To ensure consistency between the review question, inclusion criteria and data analysis, the eligibility criteria were determined following the PICOTS framework.

A standardized data extraction form was developed and piloted on a subset of five studies to ensure clarity, consistency, and comprehensiveness. After refinement, the final form was used to extract data from all eligible studies. Two reviewers (S.E.M, and M.G.S) independently performed data extraction for each study, collecting information on study characteristics (design, sample size, country), population demographics, inflammatory markers assessed, CKMS-related outcomes, follow-up duration (if applicable), effect estimates (HRs, ORs, 95% CIs), and statistical models used. Any discrepancies in extracted data were resolved through discussion between the two reviewers. If consensus could not be reached, a third reviewer (H.M) was consulted for adjudication. For studies with missing or unclear information, we attempted to contact the corresponding authors via email to obtain additional data or clarification. All decisions and correspondences were documented for transparency.

### 2.4. CKMS Stage Definitions

We operationalized CKMS stages according to the AHA advisory: Stage 0, absence of CKM risk factors; Stage 1, presence of excess or dysfunctional adiposity; Stage 2, metabolic risk factors and/or moderate- to high-risk CKD; Stage 3, subclinical CVD or CVD risk equivalents (high predicted CVD risk or very high-risk CKD); Stage 4, clinical CVD, further subdivided into Stage 4a (without kidney failure) and Stage 4b (with kidney failure). CKD was defined based on the Kidney Disease: Improving Global Outcomes (KDIGO) guidelines [16].

### 2.5. Quality Assessment

The quality of evidence and risk of bias for the included studies was evaluated by two independent assessors (T.M. and S.M.) using the Quality In Prognosis Studies (QUIPS) tool, which is a widely used, structured tool designed to assess the risk of bias in prognostic factor studies [16]. This tool assesses the quality and risk of bias in the included studies across six domains, each with specific scoring criteria: study participants (maximum scored of 7), study attrition (maximum scored of 5), prognostic factor measurement (maximum scored of 6), outcome measurement (maximum scored of 3), study confounding (maximum scored of 7), and statistical analysis and reporting (maximum scored of 4). This focused framework allows for a more relevant and precise appraisal of methodological quality in studies designed to identify or validate prognostic factors, making the QUIPS tool particularly valuable in this systematic review of prognostic literature (Appendix A). Judgements for the quality of the reported evidence were rated in accordance with the QUIPS quality of evidence assessment tool. The International Business Machine Statistical Package for the Social Sciences (IBM SPSS, version 29.0; IBM Corp., Armonk, NY, USA) was used to compute the Interrater agreement between the two independent assessors using the Intraclass Correlation Coefficient (ICC) statistical test.

#### Certainty of Evidence and Measurement Outcome

The certainty of evidence for key outcomes, including mortality, comorbidity, CKMS risk or prevalence, and mental health–inflammation pathway predictions, was evaluated using the GRADE approach in GRADEpro GDT (https://gdt.gradepro.org/app/, accessed on 15 October 2025), following standard domain-based procedures. Each outcome underwent methodological appraisal through QUIPS, followed by certainty determination in accordance with GRADE approach. The pathway from domain-level QUIPS judgments to GRADE decisions is summarized in Table 2. Two independent assessors (M.G.S. and H.M.) evaluated each GRADE domain for every outcome, resolving disagreements by discussion or third-party adjudication where needed. When applying GRADE to observational evidence, we downgraded certainty for serious indirectness (due to early CKMS operationalization and regional concentration) and, where appropriate, for imprecision. Overall, the certainty of evidence across outcomes was judged as low to moderate, with no upgrades applied.

QUIPS domain ratings for study participation, study attrition, prognostic factor measurement, outcome measurement, study confounding, and statistical analysis reporting were first summarized across studies. Domains showing moderate or high risk of bias informed corresponding GRADE downgrades in the risk of bias, indirectness, or inconsistency categories. For each GRADE domain, we determined whether there were ‘no concerns’ (no downgrade), ‘serious concerns’ (downgrade by one level, −1), or ‘very serious concerns’ (downgrade by two levels, −2). Each downgrade reduced certainty by one level.

The certainty of the evidence for mortality prediction was rated as low to moderate after being downgraded by 1 level due to concerns about indirectness. For comorbidity prediction, the certainty was also judged to be low to moderate, following downgrade by 1 level for indirectness in the available studies. When assessing CKMS risk and prevalence, we downgraded certainty by 1 level for other considerations like plausible residual confounding, which likewise resulted in a final rating of low to moderate certainty. In the case of the mental health–inflammation pathway, indirectness of the evidence led to a downgrade by 1 level, with the overall certainty again rated as low to moderate.

### 2.6. Data Synthesis

Given the heterogeneity in prognostic indices, outcome definitions, and statistical reporting, formal meta-analysis was not feasible. A random-effects analysis, using the Restricted Maximum Likelihood (REML) method on the log hazard ratio scale, was performed where incidences were poolable. For outcomes reported in single studies, we applied the Synthesis Without Meta-analysis (SWiM) approach, using vote-counting based on direction of effect and tabulated effect sizes, following SWiM guidelines [17]. Risk of bias was evaluated using QUIPS, and certainty of evidence was assessed with the GRADE framework for prognostic studies.

## 3. Results

### 3.1. Characteristics of Included Studies

Figure 1 summarizes the study selection process. In total, 1404 records were identified via database searches and other sources. After removing duplicates (*n* = 13), the titles and abstracts of 1391 studies were scanned and 461 were selected for full-text review. Of these, 13 manuscripts met the inclusion criteria and were retained in the final review. The majority of included studies were conducted in the United States (*n* = 10) [18,19,20,21,22], with three studies conducted in China (*n* = 3) [23,24]. The studies included a total of 282,016 participants, with sample sizes ranging from 6383 to 29,459. The average age of participants across studies was approximately 50 years. There were more males (*n* = 100,513) than females (*n* = 97,374) among the participants with CKMS. Most studies were prospective cohort studies (*n* = 8), while five studies used a cross-sectional approach to provide a snapshot of inflammation and CKMS-related outcomes at a single time point.

### 3.2. Quality of Evidence and Risk of Bias

Using the QUIPS quality assessment tool to assess the quality of evidence and risk of bias for the included studies, the domain study participants had excellent quality of evidence after majority (*n* = 9) of the studies scored 7 out of 7 possible scores, indicating a low risk of bias (Appendix A). Four studies showed good quality of evidence after scoring 5 (*n* = 1) and 6 (*n* = 3) points out of 7 possible scores, indicating low risk of bias. Under study attrition, majority (*n* = 7) of the included studies showed good quality of evidence and low risk of bias after scoring 4 out of 5 possible scores. Only one study reported fair quality of evidence and a moderate risk of bias after scoring 3 out of 5 possible scores. Few (*n* = 5) studies showed poor quality of evidence and a high risk of bias after scoring 0 to 2 points out of 5 possible scores. An excellent quality of evidence and a lower risk of bias was observed for prognostic factor measurement for few studies (*n* = 4) after scoring 6 out of 6 possible scores. Majority of the included studies showed good quality of evidence and low risk of bias after scoring 5 out of 6 possible scores. Outcome measurement had excellent quality of evidence and a lower risk of bias for all the included studies (*n* = 13) after scoring 3 out of 3 possible scores. Under study confounding, only two studies had excellent quality of evidence and a lower risk of bias after scoring 7 out 7 possible scores. Majority (*n* = 11) of the included studies had good quality of evidence and a low risk of bias after scoring 6 out of 7 possible scores. The statistical analysis and reporting domain indicated that majority (*n* = 11) of the included studies had excellent qualities of evidence and lower risks of bias after scoring 4 out of 4 possible scores, with only two studies showing good qualities of evidence and low risks of bias after scoring 3 out of 4 possible scores. This data was further presented using a traffic-light diagram (Appendix A).

The interrater agreement between the two independent assessors was determined using the Intraclass Correlation Coefficient (ICC). Looking at each quality assessment domain, the first assessor gave the highest rating scores for study participants with a mean (±SD) of 6.62 ± 0.650, prognostic factor measured (5.31 ± 0.480), and study confounding (6.08 ± 0.494). While the second assessor gave the highest rating scores for study attrition with a mean (±SD) of 3.69 ± 0.751, and outcome measured with a mean (±SD) of 2.54 ± 0.660. Both assessors gave similar rating scores for statistical analysis reported with a mean (±SD) of 3.77 ± 0.439. At a 95% confidence interval and a 5% margin of error, the ICC for each of the assessed domains were as follows: study participants (ICC = 0.842; *p* = 0.002), study attrition (ICC = 0.919; *p* < 0.001), prognostic factor measured (ICC = 0.900; *p* < 0.001), outcome measured (ICC = 0.873; *p* < 0.001), study confounding (ICC = 0.757; *p* = 0.006), and statistical analysis reported (ICC = 0.739; *p* = 0.017); indicative of a good rater agreement and reliable quality of evidence.

#### Certainty of Evidence and Measurement Outcome

The certainty of evidence for this systematic review was often limited by indirectness and residual confounding; several outcomes were downgraded for risk of bias due to incomplete adjustment (Appendix A). Reflecting a reasonable level of confidence but acknowledging some important limitations. The consistent finding of higher inflammation indices predicting increased mortality and comorbidities in CKMS populations was supported by multiple non-randomized studies with low risk of bias and consistent results, yet moderate certainty aroused due to serious indirectness concerns related to population and outcome variability. For CKMS risk and prevalence, the evidence was similarly moderate but tempered by limited global generalizability and potential residual confounding, indicating cautious use in broader settings. The mental health–inflammation pathway finding showed moderate certainty due to indirectness and limited data. Taken together, these moderate-certainty assessments imply that while the evidence is sufficiently robust to suggest clinical relevance and justify further research, it is not definitive for broad application without further validation. The summary of findings tables and evidence profile in provide clear documentation of study design, key domain assessments, and impact, and certainty.

### 3.3. Cross-Sectional Studies

A total of five studies (*n* = 5) employed cross-sectional designs (Appendix A), predominantly utilizing data from nationally representative databases such as NHANES [22,25,26,27]. These studies investigated the association between systemic inflammatory indices, including the DII, SII, SIRI, and leukocyte-derived ratios, and the presence or severity of CKMS. Across studies, elevated inflammatory markers were consistently associated with increased odds of CKMS or its components. For example, Zhao et al. [22] reported a 12% increase in CKMS odds per unit increase in the DII score, particularly among women over 40 years of age. Similarly, Gao et al. [23] and Asija et al. [27] found strong associations between SII/SIRI and metabolic cardiovascular renal comorbidity clusters. No prognostic interpretation was made from these designs.

### 3.4. Cohort and Longitudinal Studies

Eight studies (*n* = 8) utilized prospective or retrospective cohort designs (Table 3). Consistent evidence across these studies [18,19,21,24] demonstrates that systemic inflammation measured through markers such as leukocyte-derived ratios and indices (SIRI, SII, NLR, etc.), lipid/glucose-derived indices (TyG, hs-CRP/HDL-C, CRP–TG), and dietary inflammation indices (DII, E-DII) is significantly associated with increased mortality and progression of CKMS. Cao et al. [18] identified SIRI as a robust inflammatory marker significantly associated with increased mortality across CKMS stages. Similarly, Li et al. [2] reported that each unit increase in the CRP–TG index was associated with a 97% higher risk of all-cause mortality. Chen et al. [19] found that both the SIRI and estimated glucose disposal rate (eGDR) independently and synergistically increased mortality risk, particularly among individuals younger than under 60 years.

A study by Chen et al. [20] further illuminated the interaction between inflammation (SII), oxidative stress, and mortality risk. This was evidenced by lower OBS being strongly associated with more advanced stages of CKMS and increased all-cause and cardiovascular mortality. The OBS also mediated inflammatory effects from SII and lifestyle factors on mortality, suggesting that systemic inflammation and lifestyle-related oxidative imbalance act synergistically to worsen CKMS outcomes. Importantly, ML findings were exploratory and primarily internally assessed; calibration and external validation were limited or not reported, and therefore no claims of predictive improvement are made [30]. The prognostic value of inflammatory biomarkers such as hs-CRP/HDL-C ratio was further supported by Han et al. [24], who demonstrated that higher ratios were linearly associated with increased 10-year mortality risks in CKMS patients, especially among those with hypertension, diabetes, metabolic syndrome, or advancing age. This biomarker not only enhanced existing predictive models but also offered clinicians a practical tool for long-term risk stratification.

Zhang et al. [21] identified high-risk subgroups as older males, users of antihypertensives medications or statins, and those in CKMS stages 1 (excess adiposity) and 3 (subclinical cardiovascular disease). They also reported that TyG-derived indices, particularly when combined with anthropometric measures like waist circumference or waist-to-height ratio, were significantly correlated with both all-cause and cardiovascular mortality in CKMS individuals. These indices, reflecting inflammatory and metabolic dysregulation, further underscore the interplay between systemic inflammation, disease severity, and prognosis.

### 3.5. Synthesis Without Meta-Analysis (SWiM)

The eight prospective studies evaluated different prognostic inflammatory indices, outcome measures and statistical reporting, making meta-analysis infeasible. However, across all studies, we observed association with adverse outcomes (Appendix A). The synthesis is limited by the small number of studies per prognostic factor (one to two studies each), variability in cut-off thresholds, and differences in adjustment strategies. A SWiM approach was applied, using vote-counting based on the direction of effect as the primary synthesis method. Across the two studies reporting on SIRI there was consistency in association with increased all-cause mortality (HR 1.16–1.84), with a random-effect for completeness of (pooled HR =1.46, 95% CI 0.93–2.29, I^2^ =98.3%, τ^2^ =0.1046); however, this analysis is presented solely as exploratory and non-confirmatory. Other prognostic factors (TyG-WC, hs-CRP/HDL-C, CRP–TG Index, SII, PHQ-9 score, SIRI mediation and RAR, NPAR, SIRI, HOMA-IR) were each reported in single studies, both suggesting harmful prognostic value, but replication is lacking. Risk of bias was low to moderate in all studies due to limited sample sizes and single-center designs. Table 4: Summary of prognostic factors included in the review

## 4. Discussion

This systematic review underscores the increasing evidence that inflammation plays a central role in the development, progression, and prognosis of CKMS. Inflammatory markers like SIRI, SII, the hs-CRP/HDL-C ratio, and DII consistently demonstrated significant associations with higher all-cause and cardiovascular mortality, as well as CKMS progression. Notably, Cao et al. [18] found that individuals with advanced CKMS stages (3 or 4) and high SIRI levels (>0.81) had the greatest mortality risks, particularly among adults younger than 60 years. This is consistent with earlier research by Ridker et al. [31], which demonstrated that targeting inflammation with canakinumab, an IL-1β inhibitor, significantly reduced cardiovascular events in the CANTOS trial, independent of lipid lowering. Likewise, Ait-Oufella and Liddy [32] highlighted the crucial role of chronic low-grade inflammation in the atherogenic process, diabetes progression, and decline in renal function, all of which are important characteristics of CKMS. Collectively, the evidence supports a coherent mechanistic pathway: innate immune activation and downstream cytokine signaling drive endothelial dysfunction, promote insulin resistance, and precipitate nephron injury hallmarks that link the cardiovascular, renal, and metabolic domains of CKMS. The findings indicate that these indices have potential for hypothesis generation and prioritization for prospective validation, but standardized adjustment is required before clinical application. In practice, this could involve incorporating SIRI or SII into multivariable risk scores, applying prespecified cut points for patient triage, and evaluating whether serial changes in these indices reflect treatment response.

Additionally, cross-sectional studies provide valuable insights. Five studies, mainly using data from the nationally representative NHANES [22,25,26,27], reported strong associations between elevated dietary and leukocyte-derived inflammatory indices (DII, SII, SIRI) and higher odds of CKMS or its components. For example, Zhao et al. [22] found a 12% increase in CKMS odds for each unit rise in DII, particularly among women over 40 years, while Gao et al. [23] and Asija et al. [27] observed robust associations between SII/SIRI and cardio–metabolic–renal clusters. Although these cross-sectional findings are hypotheses-generating rather than definitive, due to the potential for reverse causation and confounding, they underscore the potential of inflammation as an early biomarker for CKMS [15,33]. To preserve causal clarity, we deliberately analyzed these studies separately from longitudinal cohorts and did not use their effect sizes to infer prognosis.

Systemic inflammation has been consistently linked to poor survival rates, a conclusion supported by this systematic review and numerous other studies. It is now recognized as a significant factor contributing to both cardiovascular and overall mortality, particularly in individuals with metabolic dysfunction and renal complications [34,35]. Chen et al. [19] and others [23] confirmed that inflammatory markers (like SIRI and SII) are particularly harmful in those with metabolic dysfunction, such as insulin resistance. Several studies [36,37,38], identified SIRI as a strong, integrative marker of chronic inflammation associated with conditions like metabolic syndrome, stroke, and other NCDs. This aligns with Hotamisligil’s findings [39], which explained how inflammation in adipose tissue disrupts insulin signaling, leading to type 2 diabetes and its complications. Our findings highlight the prognostic significance of systemic inflammation, particularly as reflected by SIRI, in predicting mortality among CKMS patients. Prioritizing patients with elevated SIRI/SII for intensified risk-factor modification (blood pressure, glycemia, LDL-C, weight management) and targeted anti-inflammatory strategies may be a pragmatic pathway for clinical translation. Furthermore, the combined effects of oxidative stress and inflammation [20], are also supported by Reuter et al. [40], who demonstrated the intricate relationship between oxidative stress, chronic inflammation, and metabolic diseases. This interconnection highlights the need to address both oxidative and inflammatory pathways to prevent or slow the progression of CKMS. Multimodal approaches that pair lifestyle (anti-inflammatory dietary patterns, physical activity) with pharmacological agents warrant prospective testing in CKMS-enriched populations.

Incorporating inflammatory indices into machine-learning models (e.g., LightGBM) shows exploratory signals of predictive utility, but we do not claim improved accuracy. [20,41]. Dietary factors, particularly pro-inflammatory diets assessed by DII, were significantly associated with CKMS outcomes [22,23]. These findings are supported by Shivappa et al. [42], who developed the DII and linked it to an increased risk of chronic diseases, including cardiovascular disease and diabetes. Their reported nonlinear, dose-dependent relationships suggest that even modest dietary changes could significantly reduce risk. Embedding DII-informed counseling into routine care pathways and evaluating its impact on intermediate inflammatory indices (e.g., SIRI/SII trajectories) could operationalize nutrition–inflammation synergy in CKMS care.

We observed heterogeneity across studies that may reflect assay platforms, biomarker cut-points, adjustment sets, and differences in CKD staging or comorbidity burden. Harmonization of definitions (including standardized thresholds for SIRI/SII and uniform outcome definitions) would improve comparability and meta-analytic power. Because most cohorts were from the U.S. and China, external validity to other regions (e.g., sub-Saharan Africa, Latin America) remains uncertain; future cohorts in diverse settings are needed to confirm transportability.

### Strengths and Limitations

This review has several strengths. By focusing on studies published between 1 January 2024 and 30 June 2025, it synthesizes the most up-to-date evidence under the newly operationalized CKMS framework. Most included studies were prospective cohorts, which strengthens temporal inference, and the large cumulative sample (*n* = 282,016) enhances precision and generalizability. We examined multiple inflammatory domains hematological indices (e.g., NLR, PLR, SII, SIRI), composite metabolic–inflammatory indices (e.g., TyG-related indices, hs-CRP/HDL-C ratio), dietary inflammation indices (e.g., DII, E-DII), and psychosocial/stress-related markers, providing a structured lens on pathways linking inflammation to CKMS risk.

A key limitation is the deliberate restriction to the post-2023 era of CKMS recognition. While this enhances internal validity by aligning studies to a common CKMS definition and reduces conceptual misclassification relative to earlier cardio-renal-metabolic constructs, it narrows the evidence base and may over-represent early-adopter regions (notably the U.S. and China). CKMS remains in early operational stages; accordingly, our findings should be interpreted as initial signals requiring replication and external validation in under-represented regions (e.g., sub-Saharan Africa, Latin America, South Asia, Eastern Europe) with attention to local diets, CKD epidemiology, and care access.

We could not perform a meta-analysis due to heterogeneity in designs, prognostic outcomes, follow-up, and marker definitions; instead, we conducted a narrative synthesis highlighting consistent associations while underscoring the need for standardized adjustment sets, harmonized staging/outcomes, and common reporting to enable future pooling and transportability assessments. Findings from machine-learning models (e.g., LightGBM) are exploratory and should be interpreted cautiously pending external validation and TRIPOD-concordant calibration reporting. Restricting to peer-reviewed publications may introduce publication and language bias. Finally, cross-sectional analyses cannot establish directionality; elevated inflammation may be a cause or a consequence of CKMS. Future longitudinal and interventional studies are needed to determine whether targeting inflammation favorably modifies CKMS trajectories.

## 5. Conclusions

Inflammatory indices (e.g., SIRI, SII, hs-CRP/HDL-C, DII) show consistent associational links with mortality and CKMS progression. Given low–to–moderate certainty, clinical translation should await standardized adjustment, TRIPOD-aligned reporting, and external validation, especially in under-represented regions. Future research should evaluate effect modification (diet quality, CKD burden, access to care) via meta-regression/stratified analyses and test whether targeting inflammation improves CKMS outcomes in prospective, well-controlled interventional studies.

## Figures and Tables

**Figure 1 ijms-27-00134-f001:**
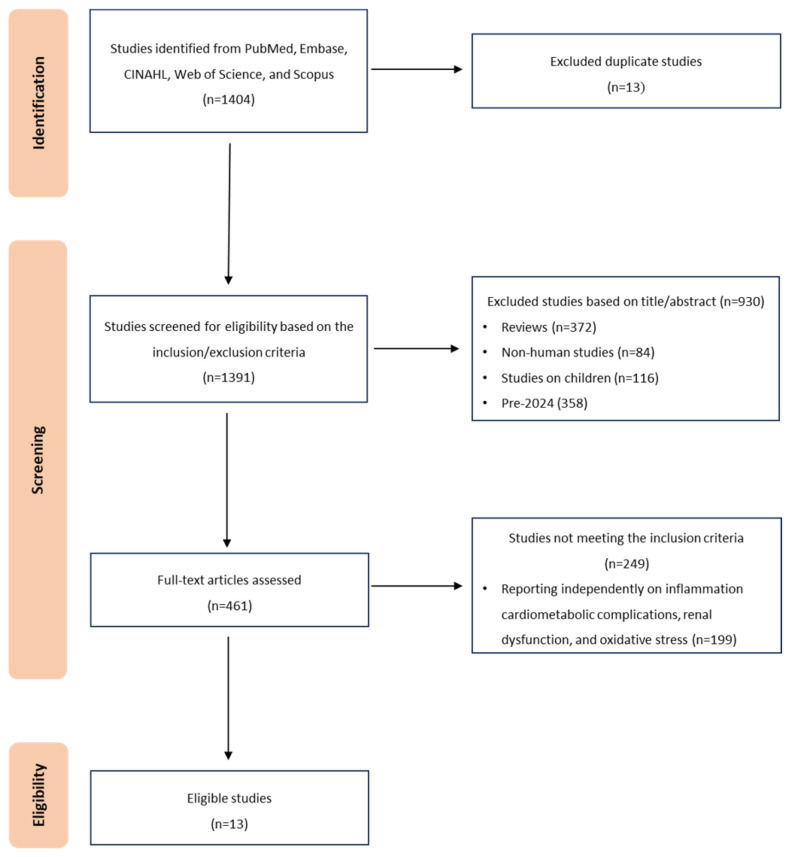
A PRISMA flow diagram depicting the study selection and inclusion process of prospective articles.

**Table 2 ijms-27-00134-t002:** GRADE Summary of findings table linking QUIPS appraisals to GRADE certainty evaluations.

Outcome	No of Studies	QUIPS Domains (Risk of Bias)	GRADE Domains						Reason for Downgrade	Final GRADE Decision	Overall Certainty
			Study Design	Risk of Bias	Inconsistency	Indirectness	Imprecision	Other			
Mortality prediction (TyG, hs-CRP/HDL-C, SIRI, SII + OBS, CRP-TG)	6	Study confounding (low risk), study participants (low risk), prognostic factor measurement (low risk), statistical analysis (low risk), study attrition (moderate risk), outcome measurement (Low risk)	Non-randomized studies	Not serious	Not serious	Serious	Not serious	None	Mortality associations were drawn mainly from US and Chinese cohorts.	Down-graded 1 level for indirect-ness	⨁⨁⨁◯ Moderate
Comorbidity prediction (SIRI	1	Study confounding (low risk), statistical analysis (low risk), study participants (low risk), prognostic factor measurement (low risk), study attrition (moderate risk), outcome measurement (low risk)	Non-randomized studies	Not serious	Not serious	Serious	Not serious	None	Single study; limited generalizability	Downgraded 1 level for indirectness	⨁⨁⨁◯ Moderate
CKMS risk/prevalence (E-DII, SII, NHR/LHR/MHR, RAR, NPAR, SIRI, HOMA-IR)	5	Study confounding (low risk), study attrition (high risk), prognostic factor measurement (low risk), outcome measurement (low risk)	Non-randomized studies	Not serious	Not serious	Not serious	Not serious	Plausible residual confounding	Residual con-founding likely	Downgraded 1 level for other considerations	⨁⨁⨁◯ Moderate
Mental Health–Inflammation pathway (PHQ-9 mediated by SIRI)	1	Confounding (low risk), statistical analysis (low risk), study confounding (low risk), study participants (low risk), prognostic factor measurement (low risk), statistical analysis (Low risk), outcome measurement (low risk)	Non-randomized studies	Not serious	Not serious	Not serious	Serious	None	Evidence from one non-randomized cohort	Downgraded 1 level for indirectness	⨁⨁⨁◯ Moderate

High certainty: ⊕⊕⊕⊕, moderate certainty: ⊕⊕⊕O, low certainty: ⊕⊕OO, very low certainty: ⊕OOO.

**Table 3 ijms-27-00134-t003:** Characteristic key findings on the included Cohort and Longitudinal Studies.

Ref.	Study Design & Setting	Population (N, Mean Age, Gender)	Inflammatory Indices	Prognostic Implications	Key Findings
Zhang et al., 2025 [21]	Prospective, USA (NHANES)	6383 participants; mean age 49.9 years; 50.5% male	TyG-derived indices	Associated with higher all-cause mortality	Elevated TyG-derived indices were predictive of all-cause mortality and cardiovascular mortality, particularly in participants with CKMS stages one and three.
Han et al., 2025 [24]	Prospective, China	6719 participants; mean age 59.0 years; 47% male	hs-CRP/HDL-C	Association observed with long-term all-cause/CVD mortality	A higher ratio was linearly associated with increased 10-year all-cause mortality, with the risk compounded by the presence of hypertension, diabetes mellitus, metabolic syndrome, and older age.
Cao et al., 2025 [18]	Prospective, USA	29,459 participants; mean age 49.8 years; 72% male	SIRI	Association with all-cause and CVD mortality reported	Individuals with a high SIRI who were in CKMS stages three or four experienced the greatest risk of mortality, especially if they were younger than 60 years.
Chen, Lian et al., 2025 [19]	Prospective, USA	18,295 participants; mean age 45.2 years; 51.6% female	SIRI + Egdr	High-risk metabolic–inflammatory phenotype	Concurrent elevation of the SIRI and a reduced eGDR doubled the risk of mortality, with a stronger association in participants younger than 60 years.
Chen, Wu et al., 2025 [20]	Prospective, USA	21,609 participants; mean age 52.0 years; 54% male	SII + Oxidative Balance Scores (OBS)	Inflammation + oxidative stress	The OBS partially mediated the association between SII and mortality; in exploratory analyses, adding OBS to a LightGBM model yielded numerically higher internal performance metrics but was not externally validated, so no claims of improved predictive performance are made.
Li et al., 2025 [28]	Prospective, CHARLS (China)	17,705 mid-to-late-adulthood participants	CRP-TG Index (CTI)	CVD incidence & mortality	Each unit increase in the CRP-TG index was associated with a 97% higher risk of death; the relationship between the index and cardiovascular disease was non-linear, whereas the association with all-cause mortality was linear.
Huang et al., 2025 [29]	Retrospective, USA (NHANES)	19,884 adults	RAR, NPAR, SIRI, HOMA-IR	CKMS prediction & mortality	Among the evaluated indices, the red blood cell distribution width-to-albumin ratio was the most associated with CKMS, with an area under the receiver operating characteristic curve of 0.907; the relationships between the indices and mortality displayed non-linear patterns.
Wang et al., 2025 [30]	Prospective, USA (NHANES)	12,314 adults	PHQ-9 score, SIRI mediation	Mental health–inflammation pathway	Higher scores on PHQ-9, reflecting depressive symptoms, were associated with increased mortality among individuals in CKMS stages one to three; the SIRI mediated approximately 12% of this relationship.

**Table 4 ijms-27-00134-t004:** Summary of prognostic factors included in the review.

Prognostic Factor	No. of Studies	Direction of Effect	Effect Size Range (HR)	Heterogeneity Summary	Risk of Bias
SIRI	2	↑ Risk	1.16–1.84	I^2^ = 98.3%, τ^2^ =0.1046, *p* < 0.0001	Low risk
SII	1	↑ Risk	1.18	Single study	Low risk
TyG-WC	1	↑ Risk	1.5	Single study	Low risk
hs-CRP/HDL-C	1	↑ Risk	1.15	Single study	Low risk
CRP–TG Index (CTI)	1	↑ Risk	1.95	Single study	Low risk
PHQ-9 score, SIRI mediation	1	↑ Risk	1.07	Single study	Low risk
RAR, NPAR, SIRI, HOMA-IR	1	↑ Risk	2.38	Single study	Moderate risk

HR: Hazard ratio. ↑ Risk: Increased Risk.

## Data Availability

No new data were created or analyzed in this study.

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
