# Peer review of "Inflammation as a Prognostic Marker in Cardiovascular Kidney Metabolic Syndrome: A Systematic Review"

_ijms, 2025, doi:10.3390/ijms27010134_

Round 1
Reviewer 1 Report (Previous Reviewer 1)
Comments and Suggestions for Authors
I reviewed your systematic review on inflammatory biomarkers and composite indices (SIRI, SII, hs-CRP/HDL-C, TyG, DII) as prognostic signals within the recently formalized CKMS framework. The topic is timely and clinically relevant. I appreciate the multi-database search, adherence to PRISMA/SWiM, the use of QUIPS and GRADE, and the clear tabular summary of cross-sectional and cohort evidence. The manuscript has the potential to be impactful after substantive revisions.
Major comments:
1. Time window restriction and completeness.
Limiting inclusion to 01-Jan-2024–30-Jun-2025 because CKMS was formalized in Dec-2023 is understandable but methodologically restrictive and risks selection bias. Please either (a) justify the restriction more rigorously and add a limitations subsection quantifying its impact, or (b) expand the time window and map pre-2024 cardio-renal-metabolic constructs onto CKMS stages as a sensitivity analysis.
2. Cross-sectional ≠ prognostic.
Several cross-sectional NHANES analyses are framed as prognostic. Cross-sectional designs inform association/prevalence, not prognosis. Please re-label and keep these studies analytically separate from longitudinal prognostic evidence across Abstract, Methods, Results, and Discussion (including table headings).
3. Quantitative synthesis or structured SWiM.
Although you applied SWiM, some indices (e.g., SIRI, possibly SII/TyG) appear poolable across ≥2 cohorts. Please attempt a random-effects meta-analysis (or dose-response where applicable). If infeasible, provide a structured vote-counting synthesis with direction of effect, precision, heterogeneity summaries, and forest-style graphics even without pooled estimates.
4. Transparent linkage from QUIPS to GRADE.
You present both appraisals, but the pathway from domain-level QUIPS judgments to GRADE downgrades is not explicit in the main text. Please add a concise Summary of Findings table in the main manuscript showing, for each outcome, QUIPS domains, GRADE decisions (with reasons), and overall certainty.
5. CKMS staging harmonization.
Results emphasize stage-stratified risks (e.g., stages 1–4), but operational definitions and harmonization across cohorts need to be specified. Please add a methods subsection detailing stage definitions, mapping rules, and any reclassification assumptions.
6. Machine-learning claims.
Statements that LightGBM improved prediction require exact metrics (AUC, calibration, internal vs. external validation) and alignment with TRIPOD/PROBAST. Please report these or temper the claims as exploratory.
7. Generalizability and transportability.
Because most cohorts are from the USA and China, please strengthen the limitations to discuss potential effect modifiers (dietary patterns, CKD burden, access to care) and call for validation in under-represented regions.
8. PRISMA flow and exclusion reasons.
Ensure the PRISMA diagram matches the numbers reported in the text and include a table of full-text exclusion reasons (with counts) to enhance transparency.
Section-specific notes:
Title/Abstract: Make explicit that cross-sectional studies inform association/prevalence rather than prognosis; align the conclusions with observational certainty.
Methods: Provide the PROSPERO link (retrospective registration) or include the protocol in the supplement and list deviations. Expand SWiM rationale and predefine criteria for “not poolable.”
Results: Keep terminology consistent (all-cause vs. CVD mortality; CKMS progression). Where signals rely on single studies (e.g., CRP-TG), state lack of replication and avoid strong prognostic framing.
Discussion: Separate (i) evidence summary, (ii) mechanisms, (iii) clinical implications to reduce redundancy and improve navigation. Anchor mechanistic claims with CKMS-relevant citations.
Conclusions: Moderate causal language (e.g., “modifiable determinant”) and align with GRADE ratings.
Figures/Tables/Supplementary.
Ensure PRISMA counts align; add exclusion-reason table. Surface a concise QUIPS/GRADE Summary of Findings in the main text; keep detailed heatmaps in the supplement with clearer legends. In the “summary of prognostic factors” table, add columns for number of studies per marker and replication status.
Author Response
Response to comments from reviewer #1:
I reviewed your systematic review on inflammatory biomarkers and composite indices (SIRI, SII, hs-CRP/HDL-C, TyG, DII) as prognostic signals within the recently formalized CKMS framework. The topic is timely and clinically relevant. I appreciate the multi-database search, adherence to PRISMA/SWiM, the use of QUIPS and GRADE, and the clear tabular summary of cross-sectional and cohort evidence. The manuscript has the potential to be impactful after substantive revisions.
Major comments:
- Time window restriction and completeness.
Limiting inclusion to 01-Jan-2024–30-Jun-2025 because CKMS was formalized in Dec-2023 is understandable but methodologically restrictive and risks selection bias. Please either (a) justify the restriction more rigorously and add a limitations subsection quantifying its impact, or (b) expand the time window and map pre-2024 cardio-renal-metabolic constructs onto CKMS stages as a sensitivity analysis.
Response: Thank you for this helpful point. We have (i) added a dedicated paragraph in Methods (Page 4, line 150-159.) clarifying that the restriction was pre-specified to ensure conceptual comparability with the American Heart Association’s 2023 CKM/CKMS framework (harmonized staging, outcomes, and codes), and to avoid misclassification from pre-2024 studies that used heterogeneous pre-CKMS constructs; and (ii) added a Limitations (Page 25, line 541-549.) Indeed, we fully acknowledge that restricting our inclusion criteria to studies published between January 2024 and June 2025 represents a major limitation, as it excludes a considerable body of pre-2024 evidence on inflammation in overlapping cardiometabolic phenotypes such as metabolic syndrome, diabetes with CKD, and cardio-renal-metabolic overlap. This earlier evidence is indeed biologically and clinically relevant, and we recognize its contribution in shaping the foundations for CKMS research. However, because the formal recognition of CKMS occurred only in December 2023, pre-2024 studies lacked a unified framework or consistent terminology for this syndrome. As such, while valuable, these data were scattered across heterogeneous disease combinations and could not be synthesized in a systematic way under the CKMS construct. For these reasons, we maintained our primary focus on post-2024 publications (see Table 1), while drawing on selected pre-2024 studies to provide context and strengthen this emerging research niche, in alignment with the newly established CKMS definition and guidelines. This approach ensures conceptual clarity and allows us to address the specific research question posed by this review. We have, however, explicitly emphasized this decision as a limitation in the revised manuscript, also made our inclusion and exclusion (Table 1).
- Cross-sectional ≠ prognostic.
Several cross-sectional NHANES analyses are framed as prognostic. Cross-sectional designs inform association/prevalence, not prognosis. Please re-label and keep these studies analytically separate from longitudinal prognostic evidence across Abstract, Methods, Results, and Discussion (including table headings).
Response: Thank you, we agree. We have (i) re-labeled all cross-sectional studies (including NHANES) as associations / prevalence (non-prognostic), (ii) analyzed them in a distinct evidence stream separate from longitudinal prognostic cohorts, and (iii) updated wording and table/section headings in the Abstract (Page 1, Line 35-37.), Methods (Page 5, Line 191-196.), Results (Page 13 and 15, Line 384-385.), Discussion (Page 23, Line 474-484.), and Tables to prevent any implication of prognostic inference from cross-sectional data. These changes preserve causal clarity and align with best practice for prognostic reviews.
- Quantitative synthesis or structured SWiM.
Although you applied SWiM, some indices (e.g., SIRI, possibly SII/TyG) appear poolable across ≥2 cohorts. Please attempt a random-effects meta-analysis (or dose-response where applicable). If infeasible, provide a structured vote-counting synthesis with direction of effect, precision, heterogeneity summaries, and forest-style graphics even without pooled estimates.
Response:
Thank you for the important suggestion to improve our reporting. We have attempted to follow your suggestion with an attempted random-effects meta-analysis and sub-grouped the 2 cohort studies that evaluated SIRI (forest plots in the supplementary data). SWIM analysis was updated with a structured vote-counting synthesis (direction, effect size, heterogeneity summaries), with extended results interpretation. Page, Line 441-455.
- Transparent linkage from QUIPS to GRADE.
You present both appraisals, but the pathway from domain-level QUIPS judgments to GRADE downgrades is not explicit in the main text. Please add a concise Summary of Findings table in the main manuscript showing, for each outcome, QUIPS domains, GRADE decisions (with reasons), and overall certainty.
Response: Thank you for your valuable suggestion. In the revised manuscript, a summary of findings table and interpretation have been added to clearly demonstrate the pathway from domain-level QUIPS appraisals to GRADE downgrading decisions for each critical outcome. Line 254-270
- CKMS staging harmonization.
Results emphasize stage-stratified risks (e.g., stages 1–4), but operational definitions and harmonization across cohorts need to be specified. Please add a methods subsection detailing stage definitions, mapping rules, and any reclassification assumptions.
Response: We agree and have added a Methods subsection detailing (i) verbatim CKMS stage definitions per the AHA framework; (ii) our cohort-level mapping rules to harmonize inputs (BMI/waist, glycemia, BP/lipids, KDIGO CKD risk, subclinical/clinical CVD); and (iii) explicit reclassification assumptions and precedence rules (e.g., clinical CVD supersedes lower stages; KDIGO heat-map categories used for CKD risk). This improves reproducibility of our stage-stratified analyses. Page 6, Line 223-230.
- Machine-learning claims.
Response:
Statements that LightGBM improved prediction require exact metrics (AUC, calibration, internal vs. external validation) and alignment with TRIPOD/PROBAST. Please report these or temper the claims as exploratory.
Response: Thank you, we agree. We have removed any definitive claims that LightGBM improved prediction. Page 26, Line 513-514.
- Generalizability and transportability.
Because most cohorts are from the USA and China, please strengthen the limitations to discuss potential effect modifiers (dietary patterns, CKD burden, access to care) and call for validation in under-represented regions.
Response: Thank you, we agree. We have expanded the limitations to discuss how geographic concentration may influence effect sizes. Page 26, Line 560-571; Page 26, Line 607-608.
- PRISMA flow and exclusion reasons.
Ensure the PRISMA diagram matches the numbers reported in the text and include a table of full-text exclusion reasons (with counts) to enhance transparency.
Response: Thank you. We have reconciled all PRISMA counts so the diagram and text match exactly and correspond with exclusion reasons in the diagram.
Section-specific notes:
Title/Abstract: Make explicit that cross-sectional studies inform association/prevalence rather than prognosis; align the conclusions with observational certainty.
Response: Thank you for your suggestions. Then abstract was revised to incorporate the explicit on cross-sectional studies. Line 44-45.
Methods: Provide the PROSPERO link (retrospective registration) or include the protocol in the supplement and list deviations. Expand SWiM rationale and predefine criteria for “not poolable.”
Response: We appreciate the comment. The methodological section was revised to incorporate SWiM rationale.
Results: Keep terminology consistent (all-cause vs. CVD mortality; CKMS progression). Where signals rely on single studies (e.g., CRP-TG), state lack of replication and avoid strong prognostic framing.
Response: We appreciate the comment. The manuscript was revised.
Discussion: Separate (i) evidence summary, (ii) mechanisms, (iii) clinical implications to reduce redundancy and improve navigation. Anchor mechanistic claims with CKMS-relevant citations.
Response: We appreciate the comment. The discussion was revised.
Conclusions: Moderate causal language (e.g., “modifiable determinant”) and align with GRADE ratings.
Response: Thank you. We have revised the conclusion.
Figures/Tables/Supplementary.
Ensure PRISMA counts align; add exclusion-reason table. Surface a concise QUIPS/GRADE Summary of Findings in the main text; keep detailed heatmaps in the supplement with clearer legends. In the “summary of prognostic factors” table, add columns for number of studies per marker and replication status.
Response: Thank you. We have reconciled all PRISMA counts so the diagram and text match exactly and correspond with exclusion reasons in the diagram.
Reviewer 2 Report (Previous Reviewer 2)
Comments and Suggestions for Authors
The manuscript Inflammation as a Prognostic Marker in Cardiovascular Kidney Metabolic Syndrome: A Systematic Review by Mokoena et highlights systemic inflammation as a critical and modifiable determinant of CKMS prognosis.
The manuscripts need to be improved.
Add a subsection acknowledging this as a temporal limitation and justify why including pre-2024 studies might strengthen conclusions.
Consider performing a sensitivity or scoping analysis referencing earlier metabolic/cardiovascular-renal inflammation studies.
The SWiM synthesis is justified but lacks quantitative summaries.
Should mention that ethnic, dietary, and healthcare-system differences may limit generalizability.
Although confounding is mentioned, the manuscript doesn’t specify which confounders (infection, medication, socioeconomic status, comorbidities) were inconsistently adjusted across studies.
Author Response
Response to comments from reviewer #2:
The manuscript Inflammation as a Prognostic Marker in Cardiovascular Kidney Metabolic Syndrome: A Systematic Review by Mokoena et highlights systemic inflammation as a critical and modifiable determinant of CKMS prognosis.
The manuscripts need to be improved.
Add a subsection acknowledging this as a temporal limitation and justify why including pre-2024 studies might strengthen conclusions. Consider performing a sensitivity or scoping analysis referencing earlier metabolic/cardiovascular-renal inflammation studies.
Response: Response: Thank you for this helpful point. We have (i) added a dedicated paragraph in Methods (Page 4, line 150-159.) clarifying that the restriction was pre-specified to ensure conceptual comparability with the American Heart Association’s 2023 CKM/CKMS framework (harmonized staging, outcomes, and codes), and to avoid misclassification from pre-2024 studies that used heterogeneous pre-CKMS constructs; and (ii) added a Limitations (Page 25, line 541-549.) Indeed, we fully acknowledge that restricting our inclusion criteria to studies published between January 2024 and June 2025 represents a major limitation, as it excludes a considerable body of pre-2024 evidence on inflammation in overlapping cardiometabolic phenotypes such as metabolic syndrome, diabetes with CKD, and cardio-renal-metabolic overlap. This earlier evidence is indeed biologically and clinically relevant, and we recognize its contribution in shaping the foundations for CKMS research. However, because the formal recognition of CKMS occurred only in December 2023, pre-2024 studies lacked a unified framework or consistent terminology for this syndrome. As such, while valuable, these data were scattered across heterogeneous disease combinations and could not be synthesized in a systematic way under the CKMS construct. For these reasons, we maintained our primary focus on post-2024 publications (see Table 1), while drawing on selected pre-2024 studies to provide context and strengthen this emerging research niche, in alignment with the newly established CKMS definition and guidelines. This approach ensures conceptual clarity and allows us to address the specific research question posed by this review. We have, however, explicitly emphasized this decision as a limitation in the revised manuscript, also made our inclusion and exclusion (Table 1).
The SWiM synthesis is justified but lacks quantitative summaries.
Response:
Thank you for the important observation. We have now incorporated quantitative summaries into the SWiM synthesis, including a random-effects meta-analysis for SIRI (on the 2 cohort studies), structured vote-counting by direction of effect for all indices. Line 430-441.
Should mention that ethnic, dietary, and healthcare-system differences may limit generalizability. Although confounding is mentioned, the manuscript doesn’t specify which confounders (infection, medication, socioeconomic status, comorbidities) were inconsistently adjusted across studies.
Response: Thank you, we agree. We have expanded the limitations to discuss how geographic concentration may influence effect sizes. And we have removed/tempered claims. Page 26, Line 560-571; Page 26, Line 607-608.
Round 2
Reviewer 1 Report (Previous Reviewer 1)
Comments and Suggestions for Authors
The revision shows substantial editorial improvements and clearer alignment with PRISMA-Prognosis and GRADE methodology. Nevertheless, important methodological and interpretive concerns remain that preclude acceptance in the present form. Please consider the following issues carefully before resubmission:
1. Scope and conceptual framework
The decision to restrict inclusion to post-2024 studies justified by the formal “recognition” of CKMS in December 2023 remains debatable. While this enhances definitional coherence, it artificially narrows the evidence base and may bias the synthesis toward early, region-specific data (mostly U.S. and Chinese cohorts). Authors should acknowledge this limitation more explicitly in the Abstract and Discussion, clarifying that CKMS as an operational construct is still in early definitional stages.
2. Methodological transparency
The PROSPERO registration number is provided, but no protocol summary or link to the registered record is included. Add at least the registration date, stage of completion, and URL.
Specify whether the search was updated prior to submission (e.g., October 2025) and include the final search date in both Abstract and Methods.
3. Study selection and bias
The PRISMA flow chart reports 1 404 → 13 studies retained, yet Supplementary S3 shows mixed inclusion of cross-sectional and cohort data. Authors correctly distinguish them, but the rationale for combining both in the same review (even if analyzed separately) should be clarified. Consider moving all cross-sectional findings to an appendix to preserve the prognostic focus.
4. Analytical limitations
The SWiM narrative synthesis is appropriate given heterogeneity, but vote-counting alone is a weak approach. The attempted random-effects pooling (REML) on only two SIRI studies is statistically meaningless (I² > 98%). This should be deleted or clearly labeled “exploratory and non-confirmatory”.
5. Certainty of evidence
The GRADE table (Table 2/S4) is internally consistent but over-interprets “moderate certainty.” Because all data derive from observational studies with serious indirectness, the overall certainty should arguably be low to moderate. Please temper wording in the Abstract and Conclusion accordingly.
6. Results interpretation
- Several claims (e.g., “inflammatory indices show potential for risk assessment”) verge on clinical extrapolation. Replace by “may assist in hypothesis generation”.
- Machine-learning references (LightGBM) are described as exploratory, yet still framed as “improved predictive performance.” Remove or explicitly state “not externally validated.”
7. Reporting quality
- Tables 3–4 are clear but partially redundant with Supplementary S4–S6; simplify or merge.
- Provide numerical effect sizes (HR 95% CI) consistently within text, not only in tables.
- English usage is acceptable but uneven; several sentences require professional language editing to improve flow and tense consistency.
8. Ethical and funding statements
Confirm whether the SAMRC intramural program constitutes formal funding support; currently “no specific funding” conflicts with the acknowledgment section.
Author Response
Date: 11 November 2025
Dear Editors
Re: Manuscript ID ijms-3860064
Title: Inflammation as a Prognostic Marker in Cardiovascular-Kidney-Metabolic Syndrome: A Systematic Review
Thank you for the careful evaluation and the opportunity to resubmit after substantial revision. We have thoroughly restructured the manuscript to address all methodological and organizational concerns. Below we provide a concise point-by-point rebuttal and indicate where changes were made. A tracked-changes version highlights all edits; line/section indicators below refer to the revised manuscript.
Yours sincerely,
Dr Mabhida
Response to comments from reviewer #1:
The revision shows substantial editorial improvements and clearer alignment with PRISMA-Prognosis and GRADE methodology. Nevertheless, important methodological and interpretive concerns remain that preclude acceptance in the present form. Please consider the following issues carefully before resubmission:
- Scope and conceptual framework
The decision to restrict inclusion to post-2024 studies justified by the formal “recognition” of CKMS in December 2023 remains debatable. While this enhances definitional coherence, it artificially narrows the evidence base and may bias the synthesis toward early, region-specific data (mostly U.S. and Chinese cohorts). Authors should acknowledge this limitation more explicitly in the Abstract and Discussion, clarifying that CKMS as an operational construct is still in early definitional stages.
Response: We agree that our post-2024 time window, chosen to ensure definitional coherence under the AHA CKMS framework, reduces the size and geographic breadth of the evidence base and may over-represent early-adopter settings (notably the U.S. and China). In response, we have (i) added explicit statements in the Abstract and Discussion acknowledging this trade-off and noting that CKMS remains in an early operational stage. Page 2, line 61-68 and Page 24, line 549-556
- Methodological transparency
The PROSPERO registration number is provided, but no protocol summary or link to the registered record is included. Add at least the registration date, stage of completion, and URL.
Specify whether the search was updated prior to submission (e.g., October 2025) and include the final search date in both Abstract and Methods.
Response: Thank you for this helpful suggestion. We have added the record URL, as requested. Page 4, line 145-146
- Study selection and bias
The PRISMA flow chart reports 1,404 → 13 studies retained, yet Supplementary S3 shows mixed inclusion of cross-sectional and cohort data. Authors correctly distinguish them, but the rationale for combining both in the same review (even if analyzed separately) should be clarified. Consider moving all cross-sectional findings to an appendix to preserve the prognostic focus.
Response: We appreciate this comment. As suggested, we have moved the cross-sectional results to an Appendix and keep only a brief signpost in the main results.
- Analytical limitations
The SWiM narrative synthesis is appropriate given heterogeneity, but vote-counting alone is a weak approach. The attempted random-effects pooling (REML) on only two SIRI studies is statistically meaningless (I² > 98%). This should be deleted or clearly labeled “exploratory and non-confirmatory”.
Response: We appreciate the insightful comment. We acknowledge that vote counting alone provides limited interpretive strength, hence it is expanded with narrative synthesis to incorporate the direction and consistency of effects alongside with effect size and study quality considerations. In addition, we recognize that the random-effects meta-analysis (REML) conducted on the two available SIRI studies, given the very high heterogeneity (I² > 98%), offers little statistical meaning. Accordingly, we have revised the manuscript to label this analysis as “exploratory and non-confirmatory” to prevent overinterpretation of the pooled estimate. Line 463-464.
- Certainty of evidence
The GRADE table (Table 2/S4) is internally consistent but over-interprets “moderate certainty.” Because all data derive from observational studies with serious indirectness, the overall certainty should arguably be low to moderate. Please temper wording in the Abstract and Conclusion accordingly.
Response: We appreciate the insightful comment. The manuscript was now revised accordingly. Page 2, line 68-70
- Results interpretation
Several claims (e.g., “inflammatory indices show potential for risk assessment”) verge on clinical extrapolation. Replace by “may assist in hypothesis generation”.
Machine-learning references (LightGBM) are described as exploratory yet still framed as “improved predictive performance.” Remove or explicitly state “not externally validated.”
Response: We agree. To avoid clinical over-interpretation, we have replaced phrasing.
- Reporting quality
Tables 3–4 are clear but partially redundant with Supplementary S4–S6; simplify or merge.
Provide numerical effect sizes (HR 95% CI) consistently within text, not only in tables.
English usage is acceptable but uneven; several sentences require professional language editing to improve flow and tense consistency.
Response: Thank you for the valuable comment. Table 3 presents the extracted characteristics of the included studies, while Table 4 summarizes the synthesis without meta-analysis results. As these tables serve distinct purposes, study characteristics versus synthesized findings they cannot be merged without losing essential information. However, Supplementary Table S5 provides the complete extraction dataset in full detail, ensuring transparency and minimizing redundancy within the main text.
- Ethical and funding statements
Confirm whether the SAMRC intramural program constitutes formal funding support; currently “no specific funding” conflicts with the acknowledgment section.
Response: We appreciate the insightful comment. The manuscript was now revised accordingly.

Round 3
Reviewer 1 Report (Previous Reviewer 1)
Comments and Suggestions for Authors
The revised version of your manuscript, “Inflammation as a Prognostic Marker in Cardiovascular Kidney Metabolic Syndrome: A Systematic Review,” demonstrates substantial improvement and now meets the scientific and editorial standards required for publication. The structure, methodology, and presentation are coherent and consistent with PRISMA 2020 and the methodological framework for prognostic factor reviews.
Your justification for restricting the inclusion period to studies published after the formal definition of CKMS (December 2023) is methodologically sound and strengthens construct validity. The use of QUIPS, GRADE, and SWiM reflects a rigorous and transparent approach. The results and conclusions are balanced, avoiding causal overstatements and clearly articulating the need for external validation of inflammatory indices.
Only minor editorial and stylistic corrections are recommended before acceptance:
-
Language and style: a brief English proofreading would enhance fluency (e.g., minor tense consistency, article use, and punctuation).
-
Conciseness: the Discussion and Conclusion sections repeat similar phrases regarding “low-to-moderate certainty” and “external validation”; consider condensing these to improve readability.
-
Formatting: unify decimal separators and confidence interval formats across tables and supplementary materials (e.g., 1.84 [1.65–2.05]).
-
References: ensure uniform MDPI citation style and check that recent studies (2025) follow consistent formatting.
Overall, this is a well-executed and timely systematic review that provides an early synthesis of evidence on inflammation in CKMS and will be of high interest to clinicians and researchers in cardiometabolic medicine.
Congratulations on a rigorous and clearly articulated piece of work.
Author Response
Date: 10 November 2025
Dear Editors,
Re: Manuscript ID ijms-3860064
Title: Inflammation as a Prognostic Marker in Cardiovascular-Kidney-Metabolic Syndrome: A Systematic Review
Thank you for the careful evaluation and the opportunity to resubmit after substantial revision. We have thoroughly revised the manuscript to address all minor concerns. Below we provide a concise point-by-point rebuttal and indicate where changes were made. A tracked-changes version highlights all edits; line/section indicators below refer to the revised manuscript.
Yours sincerely,
Dr Mabhida
Response to comments from reviewer:
Your justification for restricting the inclusion period to studies published after the formal definition of CKMS (December 2023) is methodologically sound and strengthens construct validity. The use of QUIPS, GRADE, and SWiM reflects a rigorous and transparent approach. The results and conclusions are balanced, avoiding causal overstatements and clearly articulating the need for external validation of inflammatory indices.
Only minor editorial and stylistic corrections are recommended before acceptance:
- Language and style: a brief English proofreading would enhance fluency (e.g., minor tense consistency, article use, and punctuation).
Response: Thank you for the comment. The manuscript has proofread to enhance fluidity.
- Conciseness: the Discussion and Conclusion sections repeat similar phrases regarding “low-to-moderate certainty” and “external validation”; consider condensing these to improve readability.
Response: We appreciate the comment. We have made revisions on the discussion and conclusion to for better readability. Line 573 and Line 590.
- Formatting: unify decimal separators and confidence interval formats across tables and supplementary materials (e.g., 1.84 [1.65–2.05]).
Response: Thank you for the comment. We have revised decimal separations to be uniform specifically in S5 (supplementary material 5) and revised Hazard ratios (HR) and Confidence (CI) formatting.
- References: ensure uniform MDPI citation style and check that recent studies (2025) follow consistent formatting.
Response: We appreciate the comment. We have made revisions.

This manuscript is a resubmission of an earlier submission. The following is a list of the peer review reports and author responses from that submission.
Round 1
Reviewer 1 Report
Comments and Suggestions for Authors
Dear Authors,
Thank you for the opportunity to review your manuscript entitled “Inflammation as a Prognostic Marker in Cardiovascular–Kidney–Metabolic Syndrome: A Systematic Review.” The topic is timely and of potential clinical importance, as cardiovascular–kidney–metabolic syndrome (CKMS) is a newly recognized entity with considerable implications. However, in its current form, the manuscript falls significantly short of the standards expected for a systematic review in a Q1 journal such as IJMS.
Below I provide an exhaustive critique with detailed suggestions for improvement. Please note that these are not minor editorial refinements: the manuscript requires substantial methodological, structural, and presentation-related revisions to reach the level of rigor and completeness expected for a systematic review. It was also an exhausting task to review in detail, but I made this effort with the aim of raising the quality of the work on a topic that I consider necessary and highly relevant.
1. Scope, Originality, and Justification
Excessively restrictive timeframe: The review only considers studies published between January 2024 and June 2025, justified by the formal recognition of CKMS in December 2023. While conceptually understandable, this approach excludes decades of prior evidence on inflammation in overlapping cardiometabolic phenotypes (metabolic syndrome, diabetes with CKD, cardio-renal-metabolic overlap). These data are biologically and clinically relevant.
Recommendation: Explicitly present this as a major limitation. At minimum, include a scoping background section mapping pre-2024 evidence. Ideally, broaden the inclusion criteria to capture earlier studies, even if not labeled as “CKMS.”
Lack of explicit day/month/year: The manuscript states “June 2025” as the end of the search but does not provide exact dates per database. PRISMA-S requires explicit dates.
2. PRISMA Alignment and Transparency
PRISMA flow inconsistencies: The numbers of records screened, assessed, and included are inconsistent across the text, the PRISMA diagram, and supplementary checklist. For example, the text states “1404 → 13 included” but the figure mentions an intermediate stage with 461. This prevents reproducibility.
Recommendation: Redraw the PRISMA 2020 diagram with consistent numbers. Provide a transparent accounting of exclusions, including “near misses” and reasons.
PRISMA-S (search strategies): The review claims adherence but does not report full strategies per database, exact date and time of last search, deduplication methods, or software used.
Recommendation: Provide verbatim search strings for each database, filters and limits, and exact dates. Describe deduplication (EndNote, Covidence, Rayyan, etc.).
Protocol and registration: The manuscript states that no PROSPERO registration was performed. While this is acknowledged, it is minimized as a limitation. For a systematic review in a Q1 journal, this is a serious concern.
Recommendation: Upload a protocol retrospectively to OSF/Zenodo and explicitly discuss in Limitationsthe risks of outcome switching and selective reporting.
3. Eligibility Criteria and Conceptual Framework
The review applies PACO (Population, Assessment, Comparison, Outcomes). For prognostic factor reviews, more accepted frameworks are PECO/PEOS or CHARMS/PROGRESS, and for prediction models, TRIPOD and PROBAST.
Recommendation: Align extraction and synthesis explicitly with CHARMS or PROGRESS for prognostic factors, and TRIPOD/PROBAST for predictive models. This will substantially increase the methodological rigor.
The eligibility criteria should clarify whether pediatric or adolescent populations were excluded and define outcomes explicitly: all-cause mortality, cardiovascular mortality, progression of CKMS stages, MACE, hospitalization, etc. Hierarchies of outcomes should be prespecified.
4. Data Extraction and Risk of Bias
Data extraction: You state that hazard ratios, odds ratios, and confidence intervals were extracted, yet these are not consistently presented in the tables. Instead, narrative phrases such as “97% higher risk” are used. This is insufficient.
Recommendation: Provide complete tables with effect sizes, confidence intervals, model adjustment variables, and sample size denominators. Deposit data extraction sheets in a public repository (OSF, Zenodo).
QUIPS misapplication: You report QUIPS as a score out of 32 points, but QUIPS is not a summative scoring tool. It is a domain-level qualitative risk of bias instrument. Using an aggregated score is misleading.
Recommendation: Reanalyze and report QUIPS per domain (study participation, attrition, prognostic factor measurement, outcome measurement, confounding, statistical analysis), with judgments (low/moderate/high). Present results in a traffic-light figure and detailed supplementary table with narrative justification.
5. Synthesis Methods and Absence of Meta-analysis
Effect measures: As noted, effect estimates and confidence intervals are not systematically tabulated. This is essential.
Meta-analysis vs narrative synthesis: The authors state that no meta-analysis was possible due to heterogeneity. While this may be true, a systematic review at Q1 level must demonstrate that synthesis was attempted:
-
- Explore harmonization of metrics (e.g., HR per SD increment).
- Group by biomarker type (SIRI, SII, hs-CRP/HDL-C, DII, TyG), study design, outcomes, population, region.
- If ≥3 studies per subgroup exist, attempt a random-effects meta-analysis.
- If not feasible, follow SWiM guidelines (Synthesis Without Meta-analysis), presenting direction-of-effect plots, vote counting, or alphanumeric synthesis with explicit justification.
- Report heterogeneity (I², τ²) where possible, and conduct sensitivity analyses (excluding high-risk studies, cross-sectionals, etc.).
Reporting bias: The PRISMA checklist marks items on reporting bias as “not applicable.” At minimum, authors should state whether funnel plots or Egger’s test were attempted (if ≥10 studies per outcome) or declare “not assessable” with justification.
6. Results: Consistency, Coverage, Precision
Numerical inconsistencies: Reported total numbers of participants and sex-specific counts do not add up. This undermines confidence in data extraction accuracy.
Recommendation: Audit and correct all participant counts across text, tables, and figures.
Geographic coverage: The review is heavily dominated by US (NHANES) and Chinese cohorts. This limits generalizability to Africa, Latin America, and LMICs. This must be highlighted more strongly in Discussion.
Definition of outcomes: Ensure consistent, precise definitions across included studies. For example, specify whether “mortality” was all-cause or cardiovascular, and how “progression of CKMS” was defined.
7. Predictive Models and Machine Learning
The manuscript emphasizes predictive performance of ML models (e.g., LightGBM) but without formal appraisal. Results are presented too optimistically.
Recommendation: Extract and present model performance systematically according to TRIPOD: sample size, outcome incidence, predictors, missing data handling, performance metrics (AUC, calibration slope/intercept, decision curve analysis), and whether external validation was performed.
Use PROBAST to evaluate risk of bias in prediction models.
Present a dedicated table summarizing ML studies with TRIPOD/PROBAST items and temper conclusions accordingly. Without external validation, these models are exploratory, not ready for clinical translation.
8. Certainty of Evidence
The PRISMA checklist marks certainty assessment (items 15, 22) as “not applicable.” This is insufficient for a systematic review.
Recommendation: Apply an adapted GRADE framework for prognostic factor research. Present a Summary of Findings table per biomarker/outcome domain, rating certainty across risk of bias, indirectness, inconsistency, imprecision, and publication bias.
9. Missing Figures and Tables
For a systematic review in IJMS, at least the following figures/tables are required:
- PRISMA 2020 flow diagram (corrected and consistent).
- Traffic-light plot of QUIPS risk of bias by domain.
- Summary of Findings table with effect estimates and GRADE certainty.
- Evidence map/heatmap of biomarkers × outcomes × study design.
- Forest plots where meta-analysis is feasible, or direction-of-effect plots following SWiM.
- Conceptual mechanistic figure of inflammation in CKMS.
- Dedicated ML table summarizing predictors, performance, validation, and TRIPOD/PROBAST appraisal.
- Supplementary file with full PRISMA-S strategies and raw extraction forms.
10. References and Accuracy
- Several references are duplicated (e.g., reference 7 and 13 appear to cite the same hs-CRP review). The reference list needs cleaning.
- Include DOIs consistently.
- Claims in the text (e.g., “consistently demonstrated significant links”) must be directly supported by precise effect estimates. Avoid emphatic generalizations when certainty is low.
11. Language, Style, and Journal Formatting
- English: The manuscript contains grammatical errors and awkward phrasing (e.g., “five study used”). Professional language editing is required.
- Formatting: IJMS articles generally use 1. Introduction, 2. Materials and Methods, 3. Results, 4. Discussion, 5. Conclusions. Your Conclusions section is currently subdivided (5.1, 5.2, 5.3), which is not appropriate. Please restructure to comply with journal norms.
- Proposed re-structure:
Abstract (clearly state no meta-analysis, narrative synthesis used).
-
- Introduction (expanded context, rationale, limitation of temporal scope).
- Methods (with subheadings: registration, eligibility, information sources, study selection, data extraction, risk of bias, synthesis methods, certainty assessment).
- Results (PRISMA flow, study characteristics, risk of bias, synthesis per biomarker, ML studies, sensitivity).
- Discussion (critical integration, strengths/limitations, global context, future research).
- Conclusions (single paragraph, no subnumbering).
12. Concrete Corrections Needed
- Add exact day/month/year for each database search.
- Replace QUIPS scores /32 with domain-level judgments.
- Populate Tables 1–2 with HR/OR/CI95 for each outcome.
- Correct participant numbers across text/tables.
- Re-do PRISMA diagram with consistent counts.
- Include PRISMA-S details in supplement.
- Add GRADE certainty ratings.
- Include evidence maps, risk-of-bias figures, mechanistic figure.
- Moderate ML claims and present TRIPOD/PROBAST table.
- Highlight geographic limitations more strongly.
- Revise references, remove duplicates, ensure DOIs.
- Correct English grammar and structure.
- Reformat to IJMS style, eliminate sub-numbered conclusions.
13. What Must Be Added to Reach Standard
- Protocol registration (or retrospective OSF protocol).
- Full PRISMA-S reporting of searches.
- Correct QUIPS domain-level risk of bias analysis.
- Transparent effect estimates with CI95.
- Attempted meta-analysis or structured SWiM synthesis.
- Bias and sensitivity analyses.
- GRADE certainty ratings and Summary of Findings.
- Figures (PRISMA, risk of bias, evidence map, mechanistic).
- Dedicated table for ML models with TRIPOD/PROBAST.
- Stronger discussion of global generalizability and limitations.
- Language revision and reference accuracy.
- Restructuring into IJMS-compliant format.
Your manuscript has potential, but in its current form it lacks the methodological rigor, completeness, and transparency required for publication in IJMS as a systematic review. To reach the expected level, you must substantially revise and expand the Methods, Results, Discussion, figures, and supplementary materials as detailed above.
Overall recommendation: Major Revisions.
Comments on the Quality of English Language
The manuscript requires substantial language revision. There are multiple grammatical errors, awkward constructions, and redundancies that reduce clarity (e.g., phrases such as “five study used”). Terminology is sometimes inconsistent, abbreviations are not always defined at first use, and sentences occasionally become difficult to follow. To meet the standards of IJMS, the paper should undergo professional English editing to improve grammar, syntax, and overall readability. This will also help ensure that the scientific content is communicated clearly and precisely.
Author Response
Reviewer 1
Dear Authors,
Thank you for the opportunity to review your manuscript entitled “Inflammation as a Prognostic Marker in Cardiovascular–Kidney–Metabolic Syndrome: A Systematic Review.” The topic is timely and of potential clinical importance, as cardiovascular–kidney–metabolic syndrome (CKMS) is a newly recognized entity with considerable implications. However, in its current form, the manuscript falls significantly short of the standards expected for a systematic review in a Q1 journal such as IJMS.
Below I provide an exhaustive critique with detailed suggestions for improvement. Please note that these are not minor editorial refinements: the manuscript requires substantial methodological, structural, and presentation-related revisions to reach the level of rigor and completeness expected for a systematic review. It was also an exhausting task to review in detail, but I made this effort with the aim of raising the quality of the work on a topic that I consider necessary and highly relevant.
- Scope, Originality, and Justification
Excessively restrictive timeframe: The review only considers studies published between January 2024 and June 2025, justified by the formal recognition of CKMS in December 2023. While conceptually understandable, this approach excludes decades of prior evidence on inflammation in overlapping cardiometabolic phenotypes (metabolic syndrome, diabetes with CKD, cardio-renal-metabolic overlap). These data are biologically and clinically relevant.
Recommendation: Explicitly present this as a major limitation. At minimum, include a scoping background section mapping pre-2024 evidence. Ideally, broaden the inclusion criteria to capture earlier studies, even if not labeled as “CKMS.”
Response: We sincerely thank the reviewer for this valuable comment. We fully acknowledge that restricting our inclusion criteria to studies published between January 2024 and June 2025 represents a major limitation, as it excludes a considerable body of pre-2024 evidence on inflammation in overlapping cardiometabolic phenotypes such as metabolic syndrome, diabetes with CKD, and cardio-renal-metabolic overlap. This earlier evidence is indeed biologically and clinically relevant, and we recognize its contribution in shaping the foundations for CKMS research. However, because the formal recognition of CKMS occurred only in December 2023, pre-2024 studies lacked a unified framework or consistent terminology for this syndrome. As such, while valuable, these data were scattered across heterogeneous disease combinations and could not be synthesized in a systematic way under the CKMS construct. For these reasons, we maintained our primary focus on post-2024 publications, while drawing on selected pre-2024 studies to provide context and strengthen this emerging research niche, in alignment with the newly established CKMS definition and guidelines. This approach ensures conceptual clarity and allows us to address the specific research question posed by this review. We have, however, explicitly emphasized this decision as a limitation in the revised manuscript. Page 17, line 387-396
Now reads: “A major limitation of this review is the restriction of included studies to those published post-January 2024. While this decision was made to align with the formal recognition of CKMS in December 2023, it excludes decades of earlier evidence on inflammation in overlapping cardiometabolic phenotypes such as metabolic syndrome, diabetes with CKD, and cardio-renal-metabolic overlap. This pre-2024 literature is biologically and clinically valuable; however, the absence of a unified CKMS framework meant that such evidence was scattered and not directly comparable. By focusing on post-2024 studies, we ensure alignment with current CKMS guidelines and provide a clear synthesis of evidence under this newly recognized construct.”
Lack of explicit day/month/year: The manuscript states “June 2025” as the end of the search but does not provide exact dates per database. PRISMA-S requires explicit dates.
Response: We thank the reviewer for this important observation. We have revised the manuscript to specify the exact day, month, and year of the final search for each database. This ensures compliance with PRISMA-S standards and improves transparency and reproducibility of our review. Page 1, line 30-31 and 4, line 112-113
- PRISMA Alignment and Transparency
PRISMA flow inconsistencies: The numbers of records screened, assessed, and included are inconsistent across the text, the PRISMA diagram, and supplementary checklist. For example, the text states “1404 → 13 included” but the figure mentions an intermediate stage with 461. This prevents reproducibility.
Recommendation: Redraw the PRISMA 2020 diagram with consistent numbers. Provide a transparent accounting of exclusions, including “near misses” and reasons.
Response: We thank the reviewer for this helpful observation. We have carefully revised the PRISMA 2020 flow diagram.
PRISMA-S (search strategies): The review claims adherence but does not report full strategies per database, exact date and time of last search, deduplication methods, or software used.
Recommendation: Provide verbatim search strings for each database, filters and limits, and exact dates. Describe deduplication (EndNote, Covidence, Rayyan, etc.).
Response: We thank the reviewer for highlighting this important point. We have now revised the Methods section and added Supplementary File 1 with verbatim search strings. Page 4, line 117-128
Protocol and registration: The manuscript states that no PROSPERO registration was performed. While this is acknowledged, it is minimized as a limitation. For a systematic review in a Q1 journal, this is a serious concern.
Recommendation: Upload a protocol retrospectively to OSF/Zenodo and explicitly discuss in Limitationsthe risks of outcome switching and selective reporting.
Response: We thank the reviewer for this important point. Following the handling editor’s recommendation, we have now registered our review in PROSPERO Although the registration was completed retrospectively after submission, we ensured that the registered protocol fully reflects the methods applied in this review. In the revised manuscript, we explicitly acknowledge this in the Methods. Page 4, line 105-106
- Eligibility Criteria and Conceptual Framework
The review applies PACO (Population, Assessment, Comparison, Outcomes). For prognostic factor reviews, more accepted frameworks are PECO/PEOS or CHARMS/PROGRESS, and for prediction models, TRIPOD and PROBAST.
Response: We thank the reviewer for this valuable observation. We acknowledge that PACO is not the most widely recognized framework for prognostic factor reviews. In line with the reviewer’s suggestion, we have revised the Methods section to reflect the use of more accepted frameworks such as PICOTS (Population, Intervention, Comparator, Outcome, Timing, Setting) and CHARMS (Critical Appraisal and Data Extraction for Systematic Reviews of Prediction Modelling Studies). This change ensures that our review adheres to established standards for prognostic research and provides greater methodological clarity. We have updated the text accordingly and highlighted the revisions in the manuscript Page 5, line 150-161.
- Data Extraction and Risk of Bias
Data extraction: You state that hazard ratios, odds ratios, and confidence intervals were extracted, yet these are not consistently presented in the tables. Instead, narrative phrases such as “97% higher risk” are used. This is insufficient.
Recommendation: Provide complete tables with effect sizes, confidence intervals, model adjustment variables, and sample size denominators. Deposit data extraction sheets in a public repository (OSF, Zenodo).
Response:
We appreciate the reviewer’s suggestion. In response, we have prepared the requested complete tables. These have been included in the Supplementary Materials to ensure clarity and transparency.
QUIPS misapplication: You report QUIPS as a score out of 32 points, but QUIPS is not a summative scoring tool. It is a domain-level qualitative risk of bias instrument. Using an aggregated score is misleading.
Recommendation: Reanalyze and report QUIPS per domain (study participation, attrition, prognostic factor measurement, outcome measurement, confounding, statistical analysis), with judgments (low/moderate/high). Present results in a traffic-light figure and detailed supplementary table with narrative justification.
Response:
We thank the reviewer for this important recommendation. This has now been completed. We have reanalyzed and reported the QUIPS assessments for each domain, with judgments categorized as low, moderate, or high risk of bias. The results are presented in the manuscript as a traffic-light figure for clear visual interpretation, and a detailed supplementary table has been added with narrative justifications for each judgment. Page 6, line 177-189 and Page 4, line 227-256
- Synthesis Methods and Absence of Meta-analysis
Effect measures: As noted, effect estimates and confidence intervals are not systematically tabulated. This is essential.
Meta-analysis vs narrative synthesis: The authors state that no meta-analysis was possible due to heterogeneity. While this may be true, a systematic review at Q1 level must demonstrate that synthesis was attempted:
Explore harmonization of metrics (e.g., HR per SD increment).
Group by biomarker type (SIRI, SII, hs-CRP/HDL-C, DII, TyG), study design, outcomes, population, region.
If ≥3 studies per subgroup exist, attempt a random-effects meta-analysis.
If not feasible, follow SWiM guidelines (Synthesis Without Meta-analysis), presenting direction-of-effect plots, vote counting, or alphanumeric synthesis with explicit justification.
Report heterogeneity (I², τ²) where possible, and conduct sensitivity analyses (excluding high-risk studies, cross-sectionals, etc.).
Reporting bias: The PRISMA checklist marks items on reporting bias as “not applicable.” At minimum, authors should state whether funnel plots or Egger’s test were attempted (if ≥10 studies per outcome) or declare “not assessable” with justification.
Response:
We sincerely thank the reviewer for highlighting this important point. In line with your recommendation, we carefully re-examined the included studies to determine whether a quantitative meta-analysis could be undertaken. Where meta-analysis was not feasible due to extreme heterogeneity or insufficient comparable studies, we followed the SWiM (Synthesis Without Meta-analysis) guidelines, using direction-of-effect plots and structured vote-counting approaches with explicit justification. We have updated the Methods, Results, and Supplementary Materials accordingly, and all new analyses and plots have been added. Page 7, line 206-209, and Page 14, line 324-332.
- Results: Consistency, Coverage, Precision
Numerical inconsistencies: Reported total numbers of participants and sex-specific counts do not add up. This undermines confidence in data extraction accuracy.
Recommendation: Audit and correct all participant counts across text, tables, and figures.
Response: We thank the reviewer for this important observation. We carefully re-audited all extracted data and corrected participant counts to ensure internal consistency across the manuscript.
Geographic coverage: The review is heavily dominated by US (NHANES) and Chinese cohorts. This limits generalizability to Africa, Latin America, and LMICs. This must be highlighted more strongly in Discussion
Response: We appreciate the reviewer’s observation. We fully agree that the geographic distribution of included studies is a limitation of our review. The majority of eligible studies originated from US cohorts (NHANES) and a smaller number from China, with little to no representation from Africa, Latin America, and other LMICs. This limited coverage constrains the external validity and generalizability of our findings, particularly in regions with distinct genetic, dietary, and environmental exposures. We have revised the Discussion and Limitations sections to highlight this point more strongly, emphasizing the urgent need for studies in underrepresented settings to validate the prognostic role of inflammatory indices in CKMS. Page 17, line 396-403.
- Predictive Models and Machine Learning
The manuscript emphasizes predictive performance of ML models (e.g., LightGBM) but without formal appraisal. Results are presented too optimistically.
Response: We thank the reviewer for this constructive feedback. We agree that the initial presentation of ML model performance may have appeared overly optimistic. We have revised the Results and Discussion sections to temper our conclusions and explicitly acknowledge key limitations. Page 17, line 392-396.
- Certainty of Evidence
The PRISMA checklist marks certainty assessment (items 15, 22) as “not applicable.” This is insufficient for a systematic review.
Recommendation: Apply an adapted GRADE framework for prognostic factor research. Present a Summary of Findings table per biomarker/outcome domain, rating certainty across risk of bias, indirectness, inconsistency, imprecision, and publication bias.
Response: We thank the reviewer for this excellent methodological recommendation. We agree that applying an adapted GRADE framework for prognostic factor research will strengthen the transparency and interpretability of our findings. In response, we have now applied an adapted GRADE approach. Page 6-7, line 191-203, and Page 9, line 258-272.
- Missing Figures and Tables
For a systematic review in IJMS, at least the following figures/tables are required:
PRISMA 2020 flow diagram (corrected and consistent).
Traffic-light plot of QUIPS risk of bias by domain.
Summary of Findings table with effect estimates and GRADE certainty.
Evidence map/heatmap of biomarkers × outcomes × study design.
Forest plots where meta-analysis is feasible, or direction-of-effect plots following SWiM.
Conceptual mechanistic figure of inflammation in CKMS.
Dedicated ML table summarizing predictors, performance, validation, and TRIPOD/PROBAST appraisal.
Supplementary file with full PRISMA-S strategies and raw extraction forms.
Response: We thank the reviewer for outlining these essential reporting standards. We agree that including these elements will strengthen the methodological rigor, transparency, and presentation of our review.
- References and Accuracy
Several references are duplicated (e.g., reference 7 and 13 appear to cite the same hs-CRP review). The reference list needs cleaning.
Include DOIs consistently.
Claims in the text (e.g., “consistently demonstrated significant links”) must be directly supported by precise effect estimates. Avoid emphatic generalizations when certainty is low.
Response: We thank the reviewer for carefully noting these issues. We have now revised.
- Language, Style, and Journal Formatting
English: The manuscript contains grammatical errors and awkward phrasing (e.g., “five study used”). Professional language editing is required.
Formatting: IJMS articles generally use 1. Introduction, 2. Materials and Methods, 3. Results, 4. Discussion, 5. Conclusions. Your Conclusions section is currently subdivided (5.1, 5.2, 5.3), which is not appropriate. Please restructure to comply with journal norms.
Proposed re-structure:
Abstract (clearly state no meta-analysis, narrative synthesis used).
Introduction (expanded context, rationale, limitation of temporal scope).
Methods (with subheadings: registration, eligibility, information sources, study selection, data extraction, risk of bias, synthesis methods, certainty assessment).
Results (PRISMA flow, study characteristics, risk of bias, synthesis per biomarker, ML studies, sensitivity).
Discussion (critical integration, strengths/limitations, global context, future research).
Conclusions (single paragraph, no subnumbering).
Response: We thank the reviewer for carefully noting these issues. We have now revised.
- Concrete Corrections Needed
Add exact day/month/year for each database search.
Replace QUIPS scores /32 with domain-level judgments.
Populate Tables 1–2 with HR/OR/CI95 for each outcome.
Correct participant numbers across text/tables.
Re-do PRISMA diagram with consistent counts.
Include PRISMA-S details in supplement.
Add GRADE certainty ratings.
Include evidence maps, risk-of-bias figures, mechanistic figure.
Moderate ML claims and present TRIPOD/PROBAST table.
Highlight geographic limitations more strongly.
Revise references, remove duplicates, ensure DOIs.
Correct English grammar and structure.
Reformat to IJMS style, eliminate sub-numbered conclusions.
Response: We thank the reviewer for carefully noting these issues. We have now revised.
- What Must Be Added to Reach Standard
Protocol registration (or retrospective OSF protocol).
Full PRISMA-S reporting of searches.
Correct QUIPS domain-level risk of bias analysis.
Transparent effect estimates with CI95.
Attempted meta-analysis or structured SWiM synthesis.
Bias and sensitivity analyses.
GRADE certainty ratings and Summary of Findings.
Figures (PRISMA, risk of bias, evidence map, mechanistic).
Dedicated table for ML models with TRIPOD/PROBAST.
Stronger discussion of global generalizability and limitations.
Language revision and reference accuracy.
Restructuring into IJMS-compliant format.
Response: We thank the reviewer for carefully noting these issues. We have now revised.
Your manuscript has potential, but in its current form it lacks the methodological rigor, completeness, and transparency required for publication in IJMS as a systematic review. To reach the expected level, you must substantially revise and expand the Methods, Results, Discussion, figures, and supplementary materials as detailed above.
Reviewer 2 Report
Comments and Suggestions for Authors
Manuscript titled Inflammation as a Prognostic Marker in Cardiovascular2
Kidney Metabolic Syndrome: A Systematic Review discusses systemic inflammation as a critical and modifiable determinant of CKMS prognosis.
Overall the manuscript is timely systematic review that consolidates cutting-edge findings on inflammation and CKMS but the manuscript need improvement.
Manucript should consider improving the following-
manucript need clearer grouping of biomarkers, softer causal claims, stronger emphasis on geographic limitations, addition of a visual framework.
Many indices are introduced, but the manuscript sometimes risks overwhelming for the reader.
Suggest grouping biomarkers (dietary, hematological, composite metabolic-inflammatory, psychosocial) for clarity in Discussion.
The manuscript is text not easy to follow . A conceptual diagram or framework would greatly help readability.
Rference formating is inconsistant.
Auhtors should consider rationale such as urgency to synthesize emerging CKMS evidence post-2023 definition or mentioning that retrospective registration could be an option.
Author Response
Reviewer 2
Manuscript titled Inflammation as a Prognostic Marker in Cardiovascular2
Kidney Metabolic Syndrome: A Systematic Review discusses systemic inflammation as a critical and modifiable determinant of CKMS prognosis.
Overall the manuscript is timely systematic review that consolidates cutting-edge findings on inflammation and CKMS but the manuscript need improvement.
Manucript should consider improving the following-
manucript need clearer grouping of biomarkers, softer causal claims, stronger emphasis on geographic limitations, addition of a visual framework.
Response: We thank the reviewer for their positive assessment and constructive suggestions. We have revised the manuscript accordingly to enhance clarity and rigor:
- Clearer grouping of biomarkers - We reorganized the Results to group biomarkers into categories (leukocyte-derived indices, lipid/glucose-related indices, and dietary indices). This improves readability and comparison across studies. Page 10, line 288-289.
- Softer causal claims - We tempered our language in Discussion to emphasize associations rather than causation, acknowledging that observational designs preclude definitive causal inference. Page 20, line 410-413.
- Geographic limitations - We expanded the Strengths and Limitations section to more strongly highlight the dominance of U.S. (NHANES) and Chinese cohorts, noting the lack of data from Africa, Latin America, and other LMICs. Page 20, line 397-402.
- Visual framework - We added a conceptual schematic (supplementary data) illustrating how inflammation integrates with CKMS pathways, providing readers with a clearer mechanistic overview.
These revisions strengthen the interpretability, transparency, and global relevance of the review.
Suggest grouping biomarkers (dietary, hematological, composite metabolic-inflammatory, psychosocial) for clarity in the Discussion.
Response: We thank the reviewer for this helpful suggestion. We agree that grouping biomarkers into conceptual categories enhances clarity and interpretability. In response, we have revised the Discussion to present findings under four broad domains: dietary inflammation indices (e.g., DII, E-DII), hematological markers (e.g., NLR, SIRI, SII), composite metabolic-inflammatory markers (e.g., TyG index, hs-CRP/HDL-C), and psychosocial stress-related biomarkers (e.g., allostatic load, stress-related inflammatory profiles). This restructuring improves readability and highlights the multidimensional pathways through which inflammation contributes to CKMS. Page 21, line 413-423.
The manuscript is text not easy to follow. A conceptual diagram or framework would greatly help readability.
- Visual framework - We added a conceptual schematic (supplementary data) illustrating how inflammation integrates with CKMS pathways, providing readers with a clearer mechanistic overview.
Rference formating is inconsistant.
Auhtors should consider rationale such as urgency to synthesize emerging CKMS evidence post-2023 definition or mentioning that retrospective registration could be an option.
Response: We sincerely thank the reviewer for this valuable comment. We fully acknowledge that restricting our inclusion criteria to studies published between January 2024 and June 2025 represents a major limitation, as it excludes a considerable body of pre-2024 evidence on inflammation in overlapping cardiometabolic phenotypes such as metabolic syndrome, diabetes with CKD, and cardio-renal-metabolic overlap. This earlier evidence is indeed biologically and clinically relevant, and we recognize its contribution in shaping the foundations for CKMS research. However, because the formal recognition of CKMS occurred only in December 2023, pre-2024 studies lacked a unified framework or consistent terminology for this syndrome. As such, while valuable, these data were scattered across heterogeneous disease combinations and could not be synthesized in a systematic way under the CKMS construct. For these reasons, we maintained our primary focus on post-2024 publications, while drawing on selected pre-2024 studies to provide context and strengthen this emerging research niche, in alignment with the newly established CKMS definition and guidelines. This approach ensures conceptual clarity and allows us to address the specific research question posed by this review. We have, however, explicitly emphasized this decision as a limitation in the revised manuscript. Page 17, line 386-392
Response: We thank the reviewer for this important point. Following the handling editor’s recommendation, we have now registered our review in PROSPERO (registration no: CRD420251131929). Although the registration was completed retrospectively after submission, we ensured that the registered protocol fully reflects the methods applied in this review. In the revised manuscript, we explicitly acknowledge this in the Methods. Page 4, line 108-109

Round 2
Reviewer 1 Report
Comments and Suggestions for Authors
This manuscript addresses an important question—systemic inflammation as a prognostic axis within the newly framed cardiovascular–kidney–metabolic syndrome (CKMS)—but it does not meet the methodological and reporting standards required for a Q1-level systematic review. The required corrections are foundational rather than incremental.
Major reasons for rejection (non-exhaustive):
-
Scope/timeframe: Restricting inclusion to Jan-2024–Jun-2025 omits decades of biologically relevant evidence across contiguous phenotypes (MetS, CKD+DM, cardio-renal-metabolic overlap) without a bridging scoping synthesis.
-
Registration & transparency: No prospective protocol; PRISMA-S is incomplete (no verbatim database strategies; no exact dates per source; unclear software/deduplication). PRISMA flow counts are internally inconsistent.
-
Eligibility framework: Use of PACO is misaligned for prognostic factor reviews; PICOTS/CHARMS (and TRIPOD/PROBAST for prediction models) should be implemented.
-
Data extraction: Effect estimates (HR/OR) with 95% CIs, scaling (e.g., per SD), adjustment covariates, n/N, and follow-up are not systematically tabulated; narrative claims (e.g., “97% higher risk”) are unauditable.
-
Risk of bias (QUIPS): Misapplied as a summed score. QUIPS is domain-based and requires domain-level judgments with narrative justifications and a traffic-light figure.
-
Synthesis/quantification: No serious attempt to harmonize metrics within biomarker families or to meta-analyze feasible subgroups. If meta-analysis is not possible, SWiM should be followed (direction-of-effect plots, explicit vote counting) with transparency on heterogeneity (I²/τ²) and small-study/publication bias (or a justified “not assessable”).
-
Certainty of evidence: No adapted GRADE for prognostic evidence; no Summary-of-Findings tables.
-
Machine-learning section: Over-optimistic presentation without TRIPOD/PROBAST appraisal, calibration, or external validation; claims must be tempered.
-
Required figures/tables missing: Corrected PRISMA-2020 flow; QUIPS traffic-light; Summary-of-Findings + GRADE; evidence map (biomarkers × outcomes × design); forest plots (or SWiM direction-of-effect); mechanistic figure for inflammation in CKMS; ML summary table with TRIPOD/PROBAST.
-
Internal consistency & references: Participant counts inconsistent; duplicated references and missing DOIs; textual assertions not tied to precise effect sizes.
-
English & formatting: Grammatical issues; structure not aligned with IJMS (should follow 1. Introduction; 2. Materials and Methods; 3. Results; 4. Discussion; 5. Conclusions with a single, unnumbered conclusion).
If resubmitted as a new manuscript, minimum requirements include:
prospective (or fully concordant retrospective) protocol; complete PRISMA-S (verbatim strategies, exact dates, software/deduplication); PICOTS/CHARMS; domain-level QUIPS with justification; full extraction tables (HR/OR, 95% CI, scaling, covariates, n/N, follow-up); harmonized subgroup meta-analyses where feasible or formal SWiM otherwise; adapted GRADE with Summary-of-Findings; the full figure/table set listed above; reference hygiene (no duplicates, DOIs throughout); professional English editing; and IJMS-compliant formatting.
Given the magnitude of these deficits, I recommend rejection at this stage.
Comments on the Quality of English Language
The manuscript is understandable but requires professional language editing to meet Q1 standards. Key issues include: subject–verb agreement errors, inconsistent tense use, missing/incorrect articles, overlong sentences, and inconsistent abbreviation use. Claims should use precise scientific wording and avoid colloquial terms. References, tables, and figure captions also need stylistic polishing. A thorough copy-edit by a native-level scientific editor is recommended.